# Analysis of ship emission effects on clouds over the southeastern Atlantic using geostationary satellite observations

Nikos Benas[1], Jan Fokke Meirink[1], Rob Roebeling[2], Martin Stengel[3]

[1]Royal Netherlands Meteorological Institute (KNMI), De Bilt, The Netherlands
[2]European Organisation for the Exploitation of Meteorological Satellites (EUMETSAT), Darmstadt, Germany
[3]Deutscher Wetterdienst (DWD), Offenbach, Germany

*Correspondence to*: Nikos Benas (nikos.benas@knmi.nl)

**Abstract.** This study investigates the impact of ship emissions on clouds over a shipping corridor in the southeastern Atlantic, employing geostationary-based observations, which have not been previously used in studies of this kind. Based on CLAAS-3, the 20-year (2004-2023) CLoud property dAtA set using SEVIRI, (the geostationary Spinning Enhanced Visible and InfraRed Imager), the diurnal, seasonal and long-term corridor effects on clouds are examined. Results show a significant impact of ship emissions on cloud microphysics, consistent with the Twomey effect: an increase in cloud droplet number concentration ($N_d$) and a decrease in effective radius ($r_e$). Additionally, cloud liquid water path ($W$) decreases, though changes in cloud fraction are more subtle. No clear impact on cloud optical thickness is found, implying an overall minor radiative effect of the ship emissions, although methodological limitations to detect changes in the corridor cannot be excluded. Seasonal and diurnal variations of the impact are evident, influenced by regional conditions and by the cloud thinning during the day, respectively. The long-term analysis reveals a weakening of the shipping corridor effect on $N_d$ and $r_e$ presumably following the International Maritime Organization's 2020 stricter regulations on sulfur emissions, and broader regional changes in $W$ and cloud fraction, associated with sea surface temperature variations. Focusing on a climatically important cloud regime, and including novel aspects, namely the diurnal and full seasonal cycle analyses, this study highlights the advantages and potential of geostationary satellite-based cloud observations for studying aerosol-cloud interactions.

## 1    Introduction

Clouds are a major regulator of the Earth's energy balance, modulating the amount of solar radiation reflected back to space and the amount of outgoing thermal radiation. They are also a critical component of the water cycle, redistributing water across the planet through advection. A large part of the global effect of clouds on the atmospheric radiation budget is attributed to low warm clouds, in particular stratocumulus clouds, which form extensive decks that cover large parts of the subtropical oceans. These clouds induce a cooling effect by reflecting solar radiation back to space, and a warming effect by preventing thermal radiation from escaping. Since they have a similar temperature as the underlying surface, their longwave radiative effect is rather small, and their overall impact on the Earth's climate is a cooling effect (Wood, 2012).

Since the net radiative effect of stratocumulus clouds is dominated by the shortwave component, it depends on their albedo and consequently on their microphysical properties. These, in turn, vary depending on the availability of atmospheric aerosols that act as cloud condensation nuclei. However, the ways in which aerosols interact with clouds remain highly uncertain (Bellouin et al., 2020). The instantaneous, microphysical change that fine-mode aerosols induce to liquid clouds is an increase in the cloud droplet number concentration ($N_d$) and a decrease in their effective radius ($r_e$). This process, known as the Twomey or cloud albedo effect (Twomey, 1974), occurs on a time scale of minutes to a few hours (Gryspeerdt et al., 2021). It leads to an increase in cloud albedo and consequently to an increase in radiation reflected back to space. However, subsequent adjustments can take two different paths that lead to either positive (warming) or negative (cooling) radiative effects. In the first case, the smaller droplets give rise to enhanced entrainment and evaporation, leading to less liquid water in the cloud and smaller cloud fractions, thus reducing the cloud albedo (Bretherton et al., 2007). In the second case, smaller droplets lead to suppression of precipitation and longer cloud lifetime, thus increasing the cloud fraction and albedo (Albrecht, 1989; Christensen et al., 2020). Both mechanisms have been documented and studied extensively (see e.g. the review study of Bellouin et al., 2020). They depend on the local meteorological conditions and occur on time scales of several hours (Feingold et al., 2024; Gryspeerdt et al., 2021).

In the study of aerosol interactions with clouds, ship emissions have been used extensively in the past (see e.g. Christensen et al., 2022 and references therein). They can be considered as localized sources of aerosols in otherwise undisturbed (or uniformly disturbed) environments, constituting good examples of opportunities to investigate the aerosol effects on clouds. In recent years, this research area gained momentum for two additional reasons:

1. In 2020, the International Maritime Organization (IMO) of the United Nations implemented new regulations to limit the use of sulfur in ship fuels (IMO, 2019). Since then, various studies have documented the consequences of these new regulations on ship emissions, and their effects on clouds and radiative forcing (Gettelman et al., 2024; Yuan et al., 2024; Diamond, 2023; Yuan et al., 2022; Watson-Parris et al., 2022).

2. As the effects of global warming become increasingly evident, there are also discussions to consider deliberate interventions in the climate system as a means to gain time in the effort to lower $CO_2$ emissions. One possible mechanism for this, usually called "marine cloud brightening", would be to increase the cloud albedo by injecting aerosol particles in marine clouds in order to increase their albedo and the radiation they reflect back to space (Feingold et al., 2024).

Relevant studies can be broadly categorized into campaign-based, which rely on in situ measurements from dedicated flights, and satellite-based, which use retrievals of cloud properties from satellite sensors to investigate aerosol-cloud interactions caused by ship emissions. The satellite-based approach in studying ship emission effects on clouds can be further categorized into ship track and shipping corridor studies. Individual ship tracks have been analyzed starting already with the first satellite images in the 1960s. Until recently, however, difficulties in their identification was a limiting factor in estimating their global effects on clouds (Christensen et al., 2022). This is gradually changing, with new and more sophisticated methods increasing the number of detected ship tracks and improving the relevant global estimates of these effects (Yuan et al., 2023; Manshausen

et al., 2022; Watson-Parris et al., 2022). Shipping corridor studies avoid individual ship track detection limitations by focusing on areas where ship traffic and its emissions are quasi-continuous. For example, Peters et al. (2011, 2014) examined three shipping corridors based on satellite and climate model output data. With the winds blowing mainly perpendicular to the corridors, the regions upstream and downstream of each corridor were investigated assuming clear and polluted conditions, respectively. No statistically significant effects were found, attributed to the large meteorological variability and the unknown

cloud conditions in the absence of anthropogenic emissions. Diamond et al. (2020), on the other hand, found significant effects of ship emissions on cloud $N_d$ and $r_e$, examining a corridor in the southeast (SE) Atlantic ocean, where winds typically blow parallel to the corridor and thus confine ship emissions. Examining the same corridor, Hu et al. (2021) also found significant positive effects of ship emissions on cloud albedo and fraction under low background $N_d$.

The use of geostationary data to analyze ship tracks has been so far limited to ship track identification based on reflectances

(level 1 data; Larson et al., 2022; Schreier et al., 2010). Other studies combined level 1 data from geostationary orbiters with cloud retrievals from polar orbiters (Goren and Rosenfeld, 2012; 2015). Cloud properties from geostationary satellites, based on recalibrated level 1 data, have not been previously used to analyze ship tracks. Instead, cloud properties from polar orbiting sensors have been used so far. Most studies of ship emission effects on clouds have used cloud properties retrieved from the MODerate resolution Imaging Spectroradiometer (MODIS) (e.g., Yuan et al., 2023; 2022; Diamond, 2023; Manshausen et al.,

2022; Gryspeerdt et al., 2021; Diamond et al., 2020). Contrary to geostationary-based data, the use of cloud properties from polar orbiters provides only two overpass times during the day, thus limiting the possibility of a detailed diurnal variation analysis.

In this study, we analyze the effects of ship emissions on clouds focusing on the shipping corridor that crosses the SE Atlantic ocean, connecting mainly Europe with countries in southern and SE Asia and vice versa. The suitability of this corridor in

terms of local conditions (winds typically parallel to the corridor, and an overlying persistent stratocumulus deck), for similar analyses has already been shown (Diamond et al.,2020; Hu et al., 2021; Diamond, 2023). For the first time in such a study, we use cloud properties retrieved from a geostationary sensor, the Spinning Enhanced Visible and InfraRed Imager (SEVIRI), namely the latest, third, edition of the Cloud property dAtAset using SEVIRI (CLAAS-3; Benas et al., 2023; Meirink et al., 2022), which is provided by the Satellite Application Facility on Climate Monitoring (CM SAF) of the European Organisation

for the Exploitation of Meteorological Satellites (EUMETSAT). The use of geostationary-based cloud retrievals from CLAAS-3 allows analyzing the shipping corridor effect on clouds on a range of temporal scales, including diurnal variations. Compared to polar orbiters, which typically observe a region once or twice per day, the CLAAS-3 high frequency of observations from the same sensor improves the assessment of these diurnal variations substantially. Taking advantage of the 20-year long (2004-2023) CLAAS-3 data record, we also estimate long-term changes in the corridor effect, focusing especially on the implications

of the stricter regulations implemented by IMO in 2020.

The paper is structured as follows: In Section 2 we describe the data and the methodology used in the analysis: the CLAAS-3 cloud data, the ship density data for the identification of the SE Atlantic shipping corridor, and the approaches used to identify the corridor, estimate the effects on clouds and propagate relevant uncertainties when calculating spatial and temporal averages.

Results are presented in Section 3, separated based on temporal resolution: time series averages, seasonal cycles, diurnal cycles and long term changes. A summary with main conclusions is given in Section 4.

## 2    Data and methodology

### 2.1    The CLAAS-3 cloud dataset

CLAAS-3 provides detailed information on various cloud properties, offering high spatial and temporal resolution (3 km × 3 km at nadir, every 15 minutes) and covering the period 2004 - present. As mentioned before, CLAAS-3 retrievals are based on observations from SEVIRI, which flies on board geostationary satellites Meteosat Second Generation (MSG) -1, 2, 3 and 4, covering the period 2004 – present. For the CLAAS-3 processing, SEVIRI solar channel observations were calibrated based on the latest (collection 6.1) Aqua MODIS reflectances, following the approach described in Meirink et al. (2013). Extended to include all four MSG satellites, this methodology ensures a temporal stability of CLAAS-3 that is suitable for long term analyses (see e.g. the CLAAS-3 validation report, available in Meirink et al., 2022). The instantaneous (level 2) retrievals, which are based on the calibrated (level 1) reflectances, are aggregated into (level 3) daily and monthly averages at 0.05° × 0.05° and monthly diurnal averages at 0.25° × 0.25°. The areas covered include Africa, Europe, the Atlantic Ocean and parts of South America, the Indian Ocean and the Middle East. CLAAS-3 includes cloud fractional coverage ($f_c$), cloud phase (liquid and ice), cloud top height, pressure and temperature, cloud optical thickness ($\tau$), effective radius and water path ($W$), as well as $N_d$. Details on the retrieval algorithms and the validation of CLAAS-3 are given in Benas et al. (2023) and in dedicated technical documentation available in Meirink et al. (2022).

Regarding the cloud properties examined in this study, $N_d$ is retrieved from $r_e$ and $\tau$, assuming an idealised stratiform boundary layer cloud, as described in Bennartz and Rausch (2017), while $r_e$ and $\tau$ are retrieved based on a look-up table (LUT) of pre-calculated reflectances (Roebeling et al., 2006; Nakajima and King, 1990). $W$ is also calculated from $r_e$ and $\tau$, assuming vertically homogeneous water content (e.g. Stephens, 1978). Since the 0.6 μm and the 3.9 μm channels are used for the retrieval of $r_e$ and $\tau$, it follows that $r_e$, $\tau$, $W$ and $N_d$ are available only during daytime, which is defined as solar zenith angle values lower than 75°. For consistency, $f_c$ is also provided in daytime time slots ($f_{c,\,day}$) in addition to the all-day retrievals $f_c$.

All data used in this study come from level 3 monthly averages, either of all time slots or diurnally resolved, and refer to liquid clouds only, as aggregated from level 2. Thus, the successful exclusion of ice clouds from this analysis depends on the performance of the CLAAS-3 cloud phase retrieval algorithm. Evaluation results of CLAAS-3 cloud phase, available in Meirink et al. (2022), show good performance and very good agreement with reference data sets over the wider SE Atlantic region.

Occurrences of cloud edges and broken clouds, which are associated with retrieval biases in $r_e$ and $\tau$ (and consequently $W$ and $N_d$), can also affect level 3 averages. In CLAAS-3 level 3 aggregation algorithms, level 2 results of $r_e$ and $\tau$ falling outside the LUT, which are usually associated with such cases, are excluded from level 3 averages. While this process does not ensure

complete removal of the mentioned retrieval biases, it is expected that their impact on level 3 values will be minor. Additionally, different biases in and outside the shipping corridor are not expected.

## 2.2 Identification of the shipping corridor

For the identification of the shipping corridor we use shipping traffic density data available from a collaboration of the World Bank with the IMO. This dataset is based on all hourly ship positions received by the Automatic Identification System (AIS)
between January 2015 and February 2021 (Cerdeiro et al., 2020). The data represents the total number of AIS positions reported by ships in grid cells of dimensions $0.005° \times 0.005°$ (approximately $500 \times 500$ m at the equator).

For the present study, the original ship density values are aggregated to a coarser resolution of $0.25°$ using $50 \times 50$ grid cell blocks. The threshold for ship density classification is determined by calculating the mean plus one standard deviation of the ship density values within the wider SE Atlantic region shown in Fig. 1. Using this threshold, the ship density values are
converted to a binary flag, with a flag value of 1 for cells with ship density values greater than the threshold, and 0 for cells with values below the threshold. Finally, the flag is downscaled to the original CLAAS-3 level 3 resolution of $0.05°$, in order to determine the corridor center in CLAAS-3 coordinates. This is done by selecting, in each latitudinal row, the central grid cell from the ones marked as "corridor", i.e. having a flag value of 1.

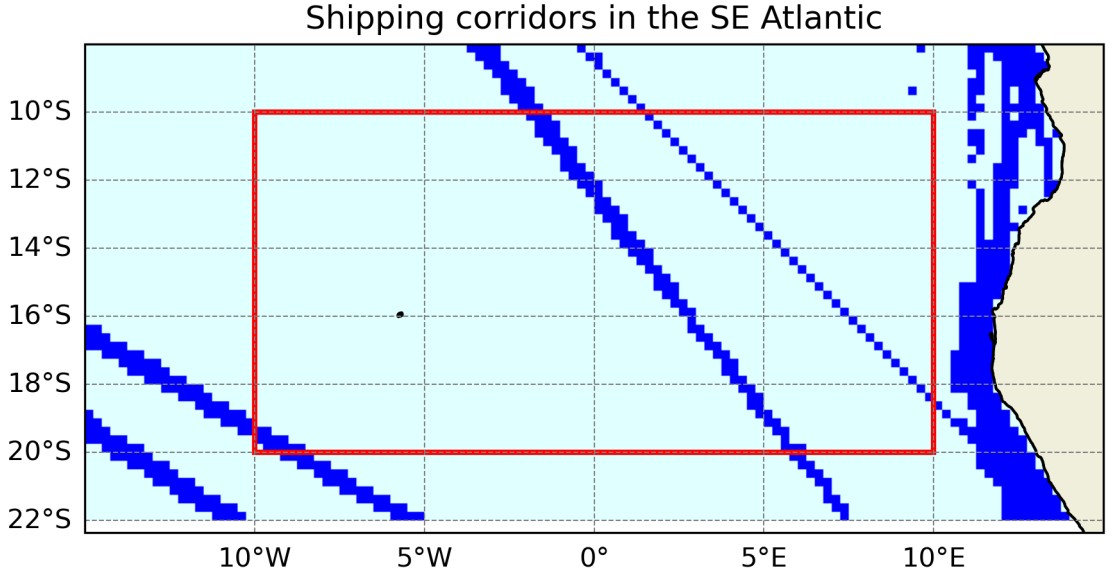

**Figure 1: Shipping corridors crossing the southeastern Atlantic, derived based on data from the Automatic Identification System and scaled to the CLAAS-3 resolution, as described in the text. The dark blue regions on the right correspond to high ship density near the African coast. The red rectangle denotes the focus region of this study.**

While the threshold used is arbitrary, it is efficient in distinguishing the major shipping corridors from lower, "background" ship density values, as Fig. 1 shows. Two corridors are identified in the study area (20° S - 10° S, 10° W - 10° E), which is

shown in red. We focus on the denser one, which produces notable differences in the $N_d$ and $r_e$ values (Figs. 2a and b). This corridor is also further away from the coast, hence least subject to the influence of terrestrial emission sources.

## 2.3 Estimation of the shipping corridor effect

The methodology used for the quantification of the shipping corridor effect on cloud properties takes advantage of the good alignment of prevailing winds in the region with the corridor orientation. This alignment constrains the emissions and renders
their effects on clouds more pronounced (Diamond, 2020).

To assess how shipping corridor emissions impact cloud properties, we first calculate the angle between the shipping corridor and the south-to-north direction. Then, for each point positioned at the corridor's center, we determine the line perpendicular to the corridor, based on this angle. We then identify the grid cells along this line and calculate their distances from the corridor center (measured in kilometers). This process yields cloud property data variations with distance from the corridor's center on
both sides. By averaging these values along the corridor line, we create a distribution centered on the midpoint of the corridor. This distribution provides insights into how cloud properties change relative to their distance from the corridor center on both sides.

The underlying assumption in this approach is that the corridor effect will manifest as a deviation from an underlying smooth distribution. Motivated by the analysis of Diamond et al. (2020), in which the shipping-affected area spans 5.0° in the east-
west direction, we define a distance of 250 km on both sides of the corridor center as a reasonable estimate of the affected area. To quantify the corridor effect, we apply a cubic fit to the 500 km affected range, based on data from the ranges 250 km - 400 km away from the corridor on both sides. These ranges, and the fitted values within the affected area, represent a scenario without the shipping corridor and they are subtracted from the actual values in order to estimate the corridor effect. When reporting an average corridor effect based on these differences, we focus on the corridor core, defined as a 150 km-wide area
(±75 km) around its center.

To assess the statistical significance of the corridor effect on a cloud variable, we examine the distribution of the corridor effect values within the corridor core region, defined above. If the 95% confidence level ($2\sigma_{std}$ range, assuming a normal distribution) does not contain zero, we conclude that the corridor effect is statistically significant at the 0.05 level.

## 2.4 Propagation of uncertainties in monthly averages

All CLAAS-3 variables come with a pixel-based estimated uncertainty at instantaneous (level 2) retrievals. This uncertainty is propagated to level 3 daily and monthly averages. The implementation of the error propagation follows Stengel et al., (2017). For the daily averages, two cases are provided in CLAAS-3, with uncertainty correlations c equal to 0.1 and 1.0. For the monthly averages, which are computed based on the daily averages, the same two scenarios are available in CLAAS-3, using the daily mean uncertainty correlation c = 0.1. For consistency, here we use the scenario where c = 0.1 also for the monthly
mean uncertainty. When averaging monthly mean data, we calculate the uncertainty of the averaged data ($\sigma_{\langle x \rangle}$) based on

$$\sigma_{\langle x \rangle}^2 = \frac{1}{N}\sigma_{std}^2 + c\langle\sigma_i\rangle^2, \tag{1}$$

where $N$ is the number of grid cells, $\sigma_{std}$ the standard deviation of the monthly mean values, and $\langle\sigma_i\rangle$ the average uncertainty of these values (based on the c = 0.1 scenario for monthly means).

For the quantification of the uncertainty of the no-ship scenarios, we repeat the cubic fit approach described above five times, varying the distance of the data ranges used in the fit from 150 km – 300 km to 350 km – 500 km. The uncertainty of the no-ship scenario is then given by the standard deviation of these five runs. As mentioned before, the range 250 km – 400 km is used for the quantification of the corridor effect. The corresponding uncertainty of the corridor effect is finally calculated as the combined uncertainty of the actual profile and the no-ship scenario.

# 3    Results

## 3.1    Average corridor effects

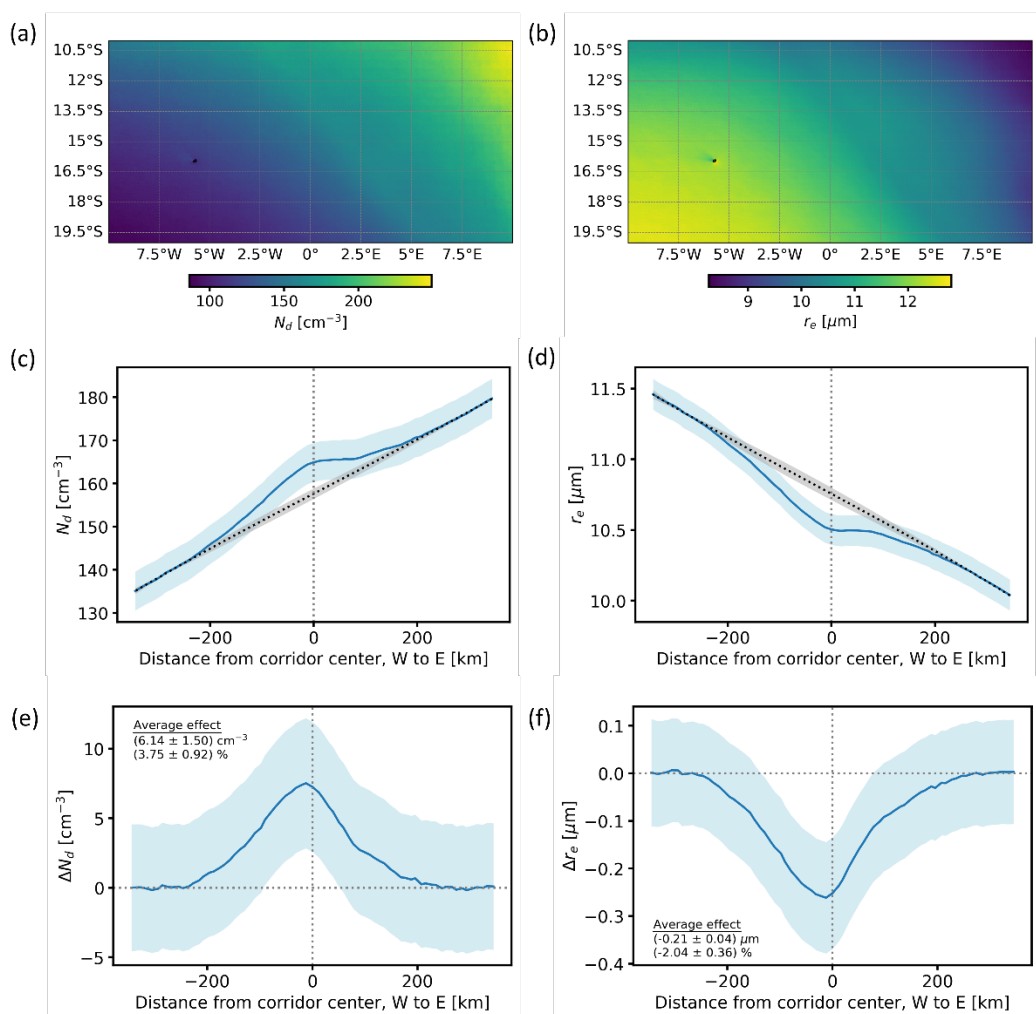

**Figure 2: Maps of monthly time series averages of (a) $N_d$, and (b) liquid $r_e$ over the study region (with corresponding uncertainties given in supplementary Fig. S1). Plots (c) and (d) show the corresponding across-corridor average distributions, with the propagated uncertainties ($1\sigma$) in light blue shades and the no-ship scenarios in dotted curves. The standard deviation of the no-ship scenarios ($1\sigma_{std}$) are shown in grey shades. Plots (e) and (f) show the corridor effect, calculated as the differences between the actual distribution and the no-ship scenarios, and their combined uncertainties. The average effects, calculated from the corridor core in absolute and relative terms, are also displayed. Dotted vertical lines denote the corridor center. In plots (e) and (f) the zero lines are also shown.**

Figures 2a and 2b show maps of $N_d$ and liquid $r_e$ averages over the study region, calculated from the entire CLAAS-3 monthly time series (2004-2023). The shipping corridor appears as a straight line in a NW-SE direction in both $N_d$ and $r_e$. Consistently with the Twomey effect, $N_d$ in the corridor is higher than in the surroundings, and $r_e$ is lower. This is shown even more clearly in the across-corridor average distributions of $N_d$ (Fig. 2c) and $r_e$ (Fig. 2d) and the ensuing differences from the no-ship scenarios (Figs. 2e and 2f, respectively). Estimation of the average corridor effect, as described in the previous section, yields

an increase in $N_d$ by 6.1±1.5 cm$^{-3}$ on average over the affected core region and a decrease in $r_e$ by 0.21±0.04 μm. The $2\sigma_{std}$
values of the distribution of the corridor effect in the corridor core are 2.3 cm$^{-3}$ and 0.08 μm, for $N_d$ and $r_e$, respectively,
showing that the effect on both is statistically significant at the 95% confidence level. The average effect values agree well
with those given by Diamond et al. (2020) for approximately the same region, even though they use a different approach and
a wider region to determine their "NoShip" scenario. For $N_d$, they found a corridor effect of ~5 cm$^{-3}$ (based only on Aqua
MODIS), while the effect found for $r_e$ was -0.29 μm for Aqua MODIS and -0.28 μm for Terra MODIS.

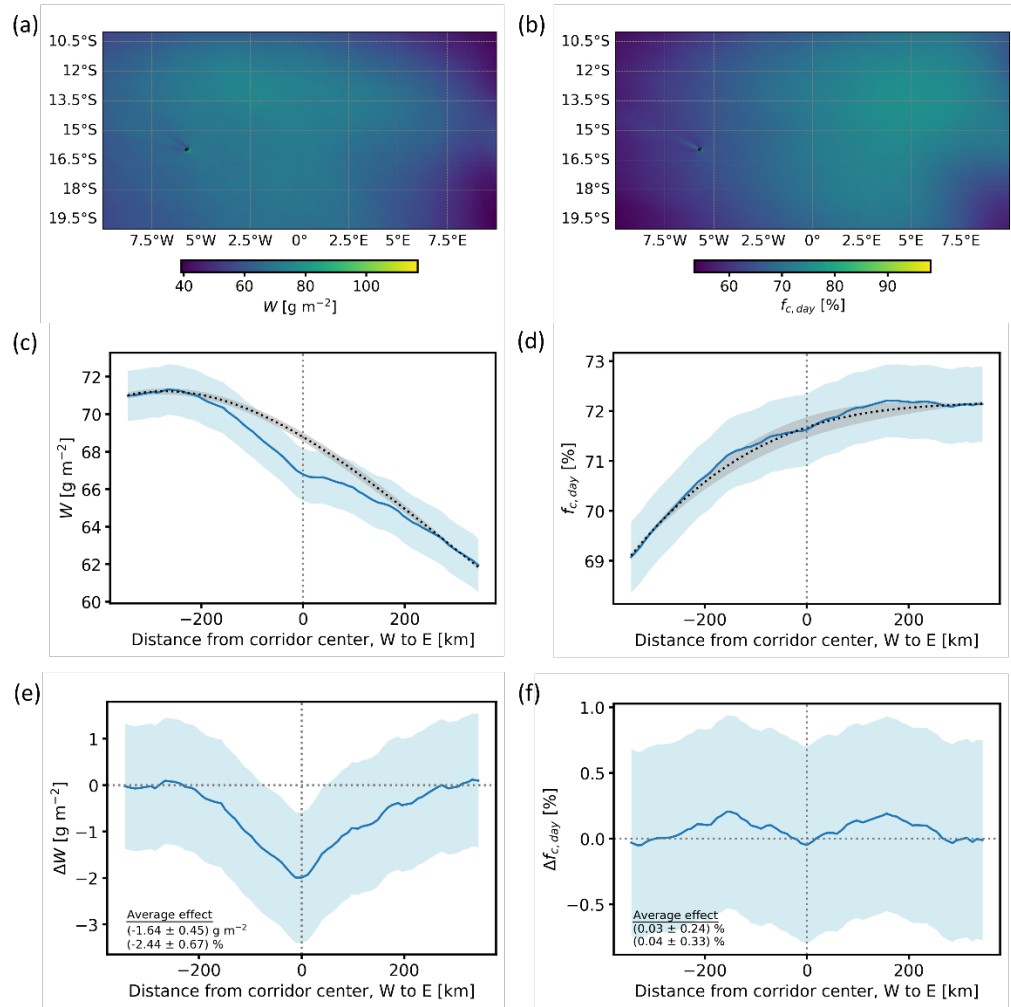

**Figure 3: As in Figure 2, but for the liquid water path ($W$) and the daytime cloud fraction ($f_{c,\,day}$).**

Corresponding results for $W$ and $f_{c,\,day}$, the two cloud variables associated with possible adjustment mechanisms, are shown in
Figure 3. In the $W$ case, the corridor perturbation is still apparent (Figs. 3c and 3e), with an effect of -1.6±0.5 g m$^{-2}$ in the core
region. The $2\sigma_{std}$ variation of the effects in the core is 0.5 g m$^{-2}$, implying that the corridor effect on $W$ is statistically significant.
The mean effect on $W$ (-1.6 g m$^{-2}$) is closer to the Aqua MODIS-based estimate of -1.3 g m$^{-2}$ from Diamond et al., (2020), than

to the estimate from Terra MODIS (-0.5 g m$^{-2}$). The response of $W$ to $N_d$ changes is complicated and depends strongly on the local meteorological and cloud conditions (see e.g. Gryspeerdt et al., 2019 and references therein). The negative response found here is consistent with studies of similar cloud regimes (Manshausen et al., 2022; Gryspeerdt et al., 2019). The higher

$N_d$ values resulting from the ship emissions lead to an increase in the entrainment rate of dry air at the cloud top, thus decreasing $W$ (Bretherton et al., 2007). Globally, however, results are not conclusive. A positive $W$ adjustment is reported in Manshausen et al. (2022), while Wall et al. (2022; 2023) find decreases in $W$ over global oceans when sulphate concentrations increase. However, these two studies are not limited to ship tracks, so their findings may not be directly comparable to the previous one. More recently, Tippett et al. (2024) showed that the $W$ response to aerosol perturbations from ship emissions is weak on

average, after correcting for biases in prior research based on tracking ship-affected air masses (Manshausen et al. 2022; 2023), related to correlations between wind and cloud properties. Several studies analysing visible ship tracks also indicate weak $W$ response to emitted aerosol (e.g., Toll et al., 2019).

In the case of $f_{c, \, day}$ there is a weak feature that appears centered along the corridor, with slightly higher values on the sides (between 100 km and 200 km) compared to the center (Fig. 3d). Estimation of the corridor effect shows a weak increase of

0.03±0.02 %, which is statistically insignificant at the 95% confidence level ($2\sigma_{std} = 0.08$). Diamond et al. (2020) also report weak, non-significant increases in daytime cloud fraction from MODIS on both Terra (0.05%) and Aqua (0.15%) over roughly the same region. However, based on the shape of the across-corridor $f_{c, \, day}$ distribution (Fig. 3d) and the corresponding effect (Fig. 3f), it is possible that the corridor emissions induce a decrease in $f_{c, \, day}$ in a narrower range; in fact, the estimated uncertainty of the no-ship scenario (grey band in Fig. 3d) includes both possibilities on the sign of the corridor effect: an

increase or a decrease in $f_{c, \, day}$. This is also shown in supplementary Fig. S2d, where the no-ship scenarios based on data from different distance ranges from the corridor center are shown: results for $N_d$, $r_e$ and $W$ show that this method is robust for calculating the corridor effect on these variables. In the $f_{c, \, day}$ case, however, if we assume that unaffected areas start closer to the corridor center and we apply the no-ship scenario fit accordingly, the corridor effect switches from positive to negative and leads to a decrease in $f_{c, \, day}$ by -0.22±0.23 % on average. While the reason why the corridor would affect $f_{c, \, day}$ in a narrower

range compared to other cloud properties is not obvious, a decrease in $f_{c, \, day}$ would be more in line with the decrease in $W$. Apart from the expected error in the method, another possible explanation of Figs. 3d and 3f is that the effect is more complicated, with a decrease closer to the corridor center and an increase on its sides. Investigation of such a scenario, however, requires modelling underlying processes beyond the scope of this study.

The same analysis was conducted also for $\tau$, the cloud optical thickness. Based on the relationship of $\tau$ with $N_d$ and $W$ (see e.g.

equation (2) in Bennartz and Rausch, 2017), and the changes over the corridor of the two latter, a small decrease should be expected in $\tau$. However, there is no apparent corridor effect in the across-corridor $\tau$ distributions, unlike the cloud properties examined before (see supplementary Fig. S3). This result may be due to $\tau$ variability across the corridor dominating the signal, and highlights a limitation in our methodology: if the corridor does not manifest as a deviation from an otherwise smooth background, no conclusive remarks on its effect can be made. Two opposing mechanisms, namely an increase in $\tau$ due to the

Twomey effect and a decrease due to the decreasing $W$, may also be acting here. Finally, this across-corridor $\tau$ distribution,

which is a time series average, does not exclude the possibility of discernible corridor effects appearing in the seasonal and diurnal results. Thus, $\tau$ is still examined in the analyses that follow.

Apart from differences in the methodology for the shipping corridor identification and the estimation of its effects, the different spatial resolutions of MODIS and SEVIRI is also a factor that may play an important role when comparing our results with those from Diamond et al. (2020), who focused on almost the same region. A coarser resolution will lead to decreased $\tau$ and increased $r_e$ values; this is the result of an increased probability of pixels containing a mixture of cloudy and clear scenes, which will lead to lower reflectances in both the visible and the shortwave infrared channel used in the retrievals. Thus, the coarser resolution of SEVIRI (~3 km) can lead to biases with corresponding MODIS results (~1 km resolution). However, it is beyond the setup of this study to disentangle such biases from differences originating in the methodology of the two studies and in the two sensors themselves. Additionally, the different resolution is not expected to affect the corridor effect significantly: the effect is estimated based on retrievals from the corridor and surrounding areas, which will be similarly affected by the coarser resolution of SEVIRI compared to MODIS.

## 3.2 Seasonal cycles

Figure 4 shows the seasonal variability in $N_d$, liquid $r_e$ (Fig. 4a), $W$ and $f_{c,\,day}$ (Fig. 4b) spatial averages over the study domain and the corresponding corridor effects. Average monthly values of the effect, along with their statistical significance at the 95% confidence level, are shown in Table 1. The corridor effect on $N_d$, $r_e$ (Fig. 4c) and $W$ (Fig. 4d) is consistent in sign and detected throughout the year, albeit with varying strength (see supplementary Fig. S4b, d and f for respective monthly profiles).

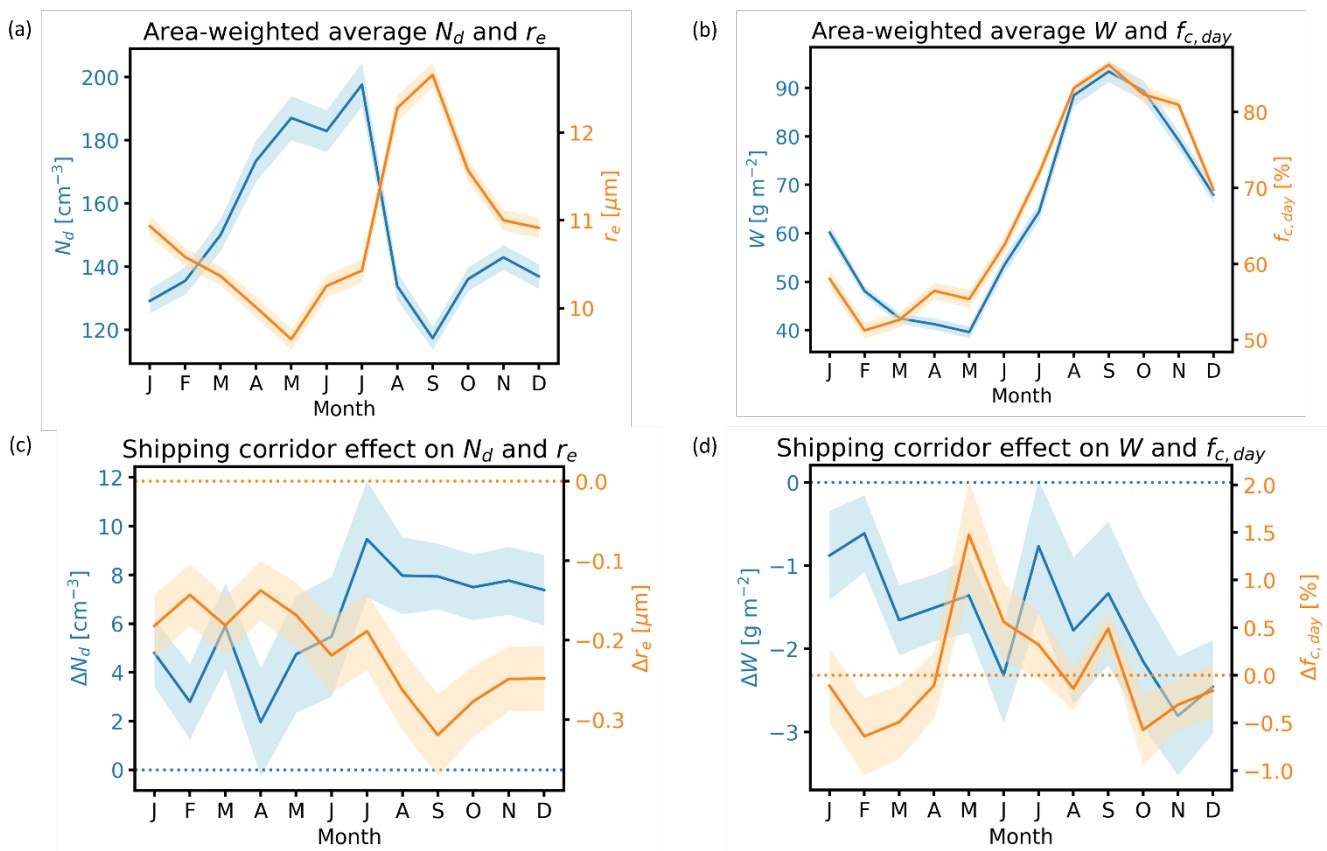

**Figure 4:** Seasonal variability of the spatially averaged (area-weighted over study domain as in Fig. 1) $N_d$ and liquid $r_e$ (a), $W$ and $f_{c, day}$ (b) and corresponding monthly corridor effects (c and d). The corridor effect values are calculated by averaging differences between the actual and the no-ship scenario values in a 150 km-wide area centered on the corridor (see also Sect. 2.3). Light-coloured bands show the uncertainties ($1\sigma$) for each parameter. Dotted lines in (c) and (d) show the zero lines for each variable.

**Table 1:** Monthly averages ($\pm 2\sigma_{std}$) of the corridor effect on $N_d$, $r_e$, $W$ and $f_{c, day}$. Statistically significant effects at the 95% confidence level are shown in bold.

|  | $N_d$ [cm$^{-3}$] | $r_e$ [μm] | $W$ [g m$^{-2}$] | $f_{c, day}$ [%] |
|---|---|---|---|---|
| January | **4.8 ± 2.0** | **-0.18 ± 0.08** | -0.9 ± 1.0 | -0.11 ± 0.16 |
| February | **2.8 ± 2.7** | **-0.14 ± 0.07** | **-0.6 ± 0.3** | **-0.64 ± 0.32** |
| March | **5.9 ± 2.3** | **-0.18 ± 0.07** | **-1.7 ± 0.6** | **-0.49 ± 0.27** |
| April | **2.0 ± 1.9** | **-0.14 ± 0.06** | **-1.5 ± 0.2** | -0.11 ± 0.31 |
| May | **4.7 ± 2.4** | **-0.17 ± 0.08** | **-1.4 ± 0.5** | **1.48 ± 0.16** |
| June | **5.5 ± 3.7** | **-0.22 ± 0.08** | **-2.3 ± 0.6** | **0.56 ± 0.28** |
| July | **9.5 ± 1.4** | **-0.19 ± 0.09** | **-0.8 ± 0.2** | **0.32 ± 0.30** |
| August | **8.0 ± 1.8** | **-0.26 ± 0.11** | **-1.8 ± 0.8** | **-0.14 ± 0.11** |

| | | | | |
|---|---|---|---|---|
| September | 7.9 ± 2.5 | -0.32 ± 0.11 | -1.3 ± 1.1 | 0.49 ± 0.10 |
| October | 7.5 ± 2.6 | -0.28 ± 0.10 | -2.1 ± 0.7 | -0.57 ± 0.16 |
| November | 7.8 ± 2.3 | -0.25 ± 0.08 | -2.8 ± 0.6 | -0.31 ± 0.43 |
| December | 7.4 ± 3.4 | -0.25 ± 0.10 | -2.5 ± 0.8 | -0.16 ± 0.25 |

The highest $N_d$ values appear in May, June and July, and secondarily in March and April (Fig. 4a). Lower values occur in local spring and summer, with the lowest occurring in September. Liquid $r_e$ is anti-correlated with $N_d$, with the highest values occurring in August and September and the lowest in April-June. The seasonal cycle of $N_d$ is very similar to corresponding results from previous studies based on MODIS (Bennartz and Rausch, 2017, Grosvenor et al., 2018), with CLAAS-3 exhibiting higher values. The peak in July and the rapid drop in August and September are prominent in all data sets. A similar $N_d$ seasonal cycle peaking in summer is also found in Li et al., (2018) over a larger region and using both MODIS and CALIPSO data. Zeng et al. (2014) also find $N_d$ peaks west of Namibia in April and July, based on MODIS and CALIOP data. Previous studies report that the seasonal cycle of $N_d$ in this region is affected mainly by aerosol emissions from southern Africa (Grosvenor et al., 2018; Li et al., 2018; Bennartz and Rausch, 2017). The presence of absorbing aerosols from biomass burning activities above the cloud layer, which is typical in this region for certain months, could also have an artificial effect on the retrieved $\tau$ and $r_e$: by darkening the scene, especially in the 0.6 μm channel, smoke aerosols over clouds can lead to the retrieval of lower $\tau$ as well as lower $r_e$. However, when using the 3.9 μm channel reflectance, the effect on $r_e$ is – in comparison to shorter wavelengths - relatively small (Haywood et al., 2004). Since the $N_d$ retrieval depends weakly on $\tau$ and much more strongly on $r_e$ (see e.g. equation (2) in Bennartz and Rausch, 2017), the absorbing aerosol effect on $N_d$ is also expected to be modest compared to retrievals based on smaller wavelengths (e.g., 1.6 μm). Furthermore, smoke aerosols in the region are expected to have a similar effect in both corridor-affected and unaffected parts, thus influencing the corridor effect calculations even less than area-wide average values.

The strong seasonal cycle of the (in-cloud) $W$ follows that of $f_{c,\,day}$ (Fig. 4b), which peaks between August and November. In turn, the seasonal peak in $f_{c,\,day}$ coincides with the season of maximum static stability, which follows inversely the seasonal cycle of sea surface temperature (SST; Klein and Hartmann, 1993). Thus, thicker clouds in September-November can explain the peak in $W$, but also the higher $r_e$ values, which are expected to grow with height. This season is also reported by Diamond et al., (2020) as giving the strongest corridor signal. In our case, the corridor effect on $N_d$ is consistently positive and statistically significant throughout the year (meaning more cloud droplets in observations compared to a scenario without ships) and appears stronger from July to December (Fig. 4c and Table 1). Similarly, for $r_e$, with a consistently significant negative corridor effect, meaning smaller cloud particles in reality compared to a no-ship scenario. The presence of the corridor causes a decrease in $W$ also throughout the year (Fig. 4d and Table 1). In the case of $f_{c,\,day}$, the corridor signal is ambiguous and appears to change sign during the year (Fig. 4d and Table 1). However, a closer examination of monthly across-corridor profiles of effects on $f_{c,\,day}$ (supplementary Fig. S4h) shows that only in austral summer months (mainly October, December, February and March) do the profiles appear as deviations roughly symmetrical across the corridor centre, indicating a distinguishable corridor effect.

In these cases, actual values are lower than in the no-ship scenario. Thus, while the time series average corridor effect on $f_{c, day}$ is ambiguous (Fig. 3f), a clearer signal of $f_{c, day}$ decreasing over the corridor appears in austral summer. In other months, it is difficult to draw any conclusion on the shipping corridor effect on $f_{c, day}$. Therefore, a statistically significant effect in these months (i.e. May – September values in Table 1), which is not accompanied by an across-corridor profile of changes as described before, should not be interpreted as a meaningful corridor effect.

Results of across-corridor $\tau$ profiles per month are shown in supplementary Fig. S4i and corresponding profiles of corridor effect in Fig. S4j. The latter, however, are only provided for completeness: the corridor does not manifest convincingly in any month examined, so the calculated effects in Fig. S4j cannot be used to draw any conclusion.

## 3.3    Diurnal cycles

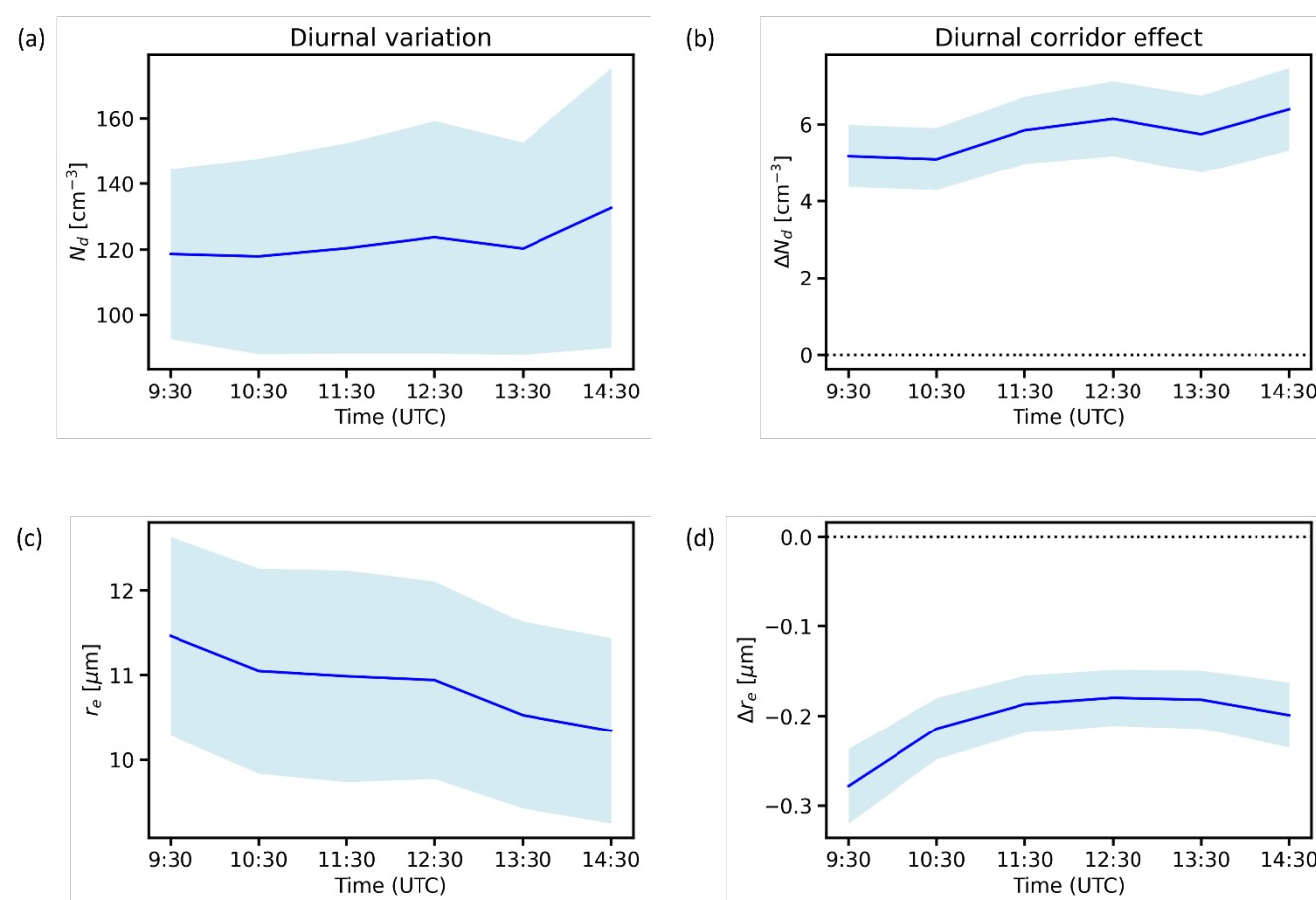

**Figure 5: Diurnal variability of the spatially averaged (area-weighted) $N_d$ (a) and liquid $r_e$ (c) and corresponding diurnal corridor effects (b and d). Light blue bands show the standard deviation for each parameter.**

Figure 5 shows the diurnal variability of the $N_d$ and $r_e$ averages over the study area (Figs. 5a, c), and the corresponding average corridor effects (Figs. 5b, d). Corresponding across-corridor profiles of effects for every time slot when data are available are

shown in the supplementary Fig. S5. Since $N_d$ and $r_e$ can be retrieved only during the day, it was required that for each time slot, the study region is completely covered with data from all months in the time series. In this way, time slots at the beginning and end of the day, when high solar zenith angles may lead to problematic retrievals, were omitted. For the same reason, the two extreme of the remaining available time slots (0800 UTC and 1500 UTC) were also removed from the analysis. A statistical significance analysis of the diurnal cycles was not performed, due to the coarser resolution of these data, which leads to very

few points inside the corridor core.

$N_d$ is nearly constant during the available time slots of the day (Fig. 5a). The effect of the corridor on $N_d$ also appears stable during the day, ranging between 5 cm$^{-3}$ and 6 cm$^{-3}$, with somewhat higher values in the afternoon (Fig. 5b). Liquid $r_e$ decreases slightly during the day (Fig. 5c), in consistence with previous findings (Seethala et al., 2018).  The corridor effect on $r_e$ also gets weaker as the day progresses, stabilizing early in the afternoon (Fig. 5d).

Diamond et al. (2020) report larger corridor effects on $N_d$ during the Terra overpass (~10:30 local time), compared to Aqua (~13:30 local time), contrary to this study. Their estimation of the relative change in Terra $N_d$ is based on respective changes in $W$ and $r_e$, because of the absence of a published Terra $N_d$ product. For $r_e$ they report similar effects during the Terra and Aqua overpasses, as also found here (Fig. 5d).

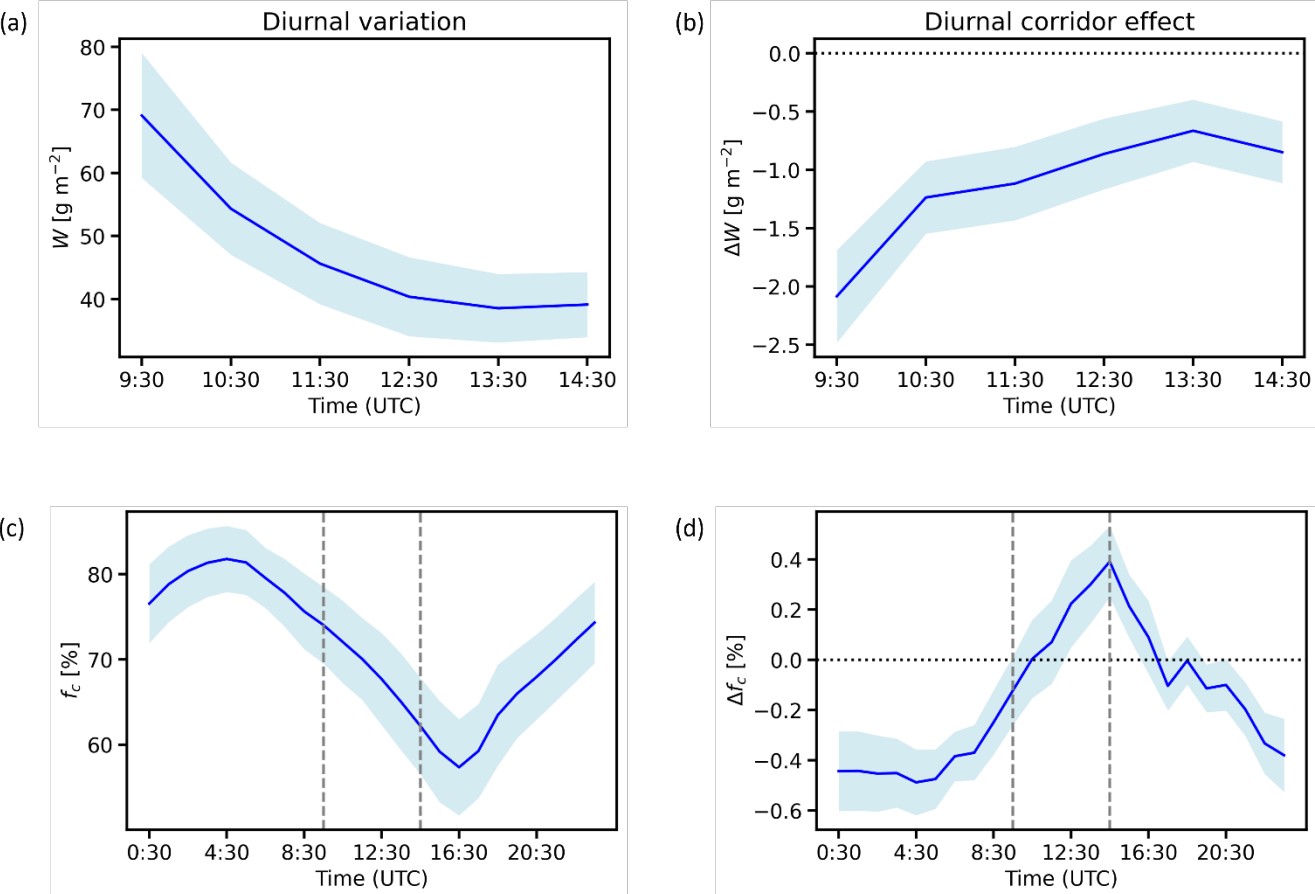

**Figure 6: As in Figure 5 but for $W$ and $f_c$. The two vertical dashed lines in the diurnal variation of $f_c$ (c) and corresponding corridor effect (d) represent the start and end time of the day when $N_d$, $r_e$ and $W$ retrievals are also available.**

Corresponding results for $W$ show a strong decrease during the day, accompanied by a weakening of the corridor effect (Figs. 6a and b). In the afternoon, the corridor effect on $W$ is about half compared to the morning value. The decrease in $W$ during the day is in good agreement with results of Seethala et al. (2018): they also report peak values of $W$ early in the morning and

340 a decrease until the afternoon. A small decrease is also reported for $r_e$ during the day. The decrease in $W$ is associated with the cloud thinning, which is apparent in the $f_c$ diurnal plot (Fig. 6c).

The estimated corridor effect on $W$ by Diamond et al. (2020) is lower in the morning (Terra overpass) than in the afternoon (Aqua overpass). This contrasts with our findings, which show similar decreases in the two relevant time slots (10:30 UTC and 13:30 UTC in Fig. 6b), with a somewhat weaker effect during the Aqua overpass. Other studies of ship tracks in the north

eastern Pacific, where the cloud conditions are similar, show even larger $W$ decreases for Terra MODIS than for Aqua MODIS, with the former being almost double the latter (Christensen et al., 2009; Segrin et al., 2007). These results hinder drawing concrete conclusions regarding the differences in the corridor effect between morning and afternoon, and highlight the value of using observations in more frequent time intervals, such as CLAAS-3, for similar analyses. However, diurnal observations

should also be treated with caution, since they may suffer from diurnally varying biases related to scattering geometry (Smalley and Lebsock, 2023).

Since $f_c$ can be retrieved also at night time, the full 24-hour cycle variation is available, showing the cloud thinning during day and the thickening during night (Fig. 6c). It is interesting to note that the corridor effect on $f_c$ changes sign between day and night (Fig. 6d). It is negative during the night and becomes positive at day, contrary to what would be expected from the diurnal cycles of precipitation and evaporation, which maximize at night and day, respectively (e.g. Sandu et al., 2008). A closer examination of the $f_c$ difference profiles per time slot, however, shows that the positive day time differences have a local minimum at the corridor center (supplementary Fig. S5d). This is reminiscent of the pattern found also in the seasonal analysis of $f_{c, day}$, suggesting that selection of a narrower range for the corridor-affected area during the day would lead to lower (or even negative) effects on $f_c$. However, since this indication of a narrower affected area does not appear in other variables, no concrete conclusion can be drawn regarding the effect of ship emissions on $f_c$ during the day. It can be safely concluded, however, that during night the shipping corridor exerts a negative effect on $f_c$, as can be shown by examining shipping corridor and no-ship scenario profiles of $f_c$ on an individual time slot basis (supplementary Fig. S6).

Results on the diurnal variation of $\tau$ over the region, across-corridor profiles and corresponding estimates of corridor effect are shown in supplementary Fig. S7. Average values of $\tau$ decrease during the day, as expected (Fig. S7a). The diurnal variation of the corridor effect shows practically no effect in the morning, with slightly positive values in the afternoon (Fig. S7b). However, these results should be interpreted with caution, since corresponding across-corridor profiles have high uncertainty and show little or no appearance of corridor-centered perturbations (Figs. S7c and S7d-i).

## 3.4    Long term changes

In order to examine the strength of the corridor effect before and after 2020, when the stricter IMO regulations on sulfur-containing emissions were implemented, we repeated the analysis described in Sect. 3.1 for two time periods: 2004-2019 and 2020-2023. While the two time periods are not equal, the longer period before 2020 is more representative of the entire CLAAS-3 time range, and reduces the risk of being affected by large year-to-year variability. The across-corridor distributions of the effect on $N_d$, $r_e$, $W$ and $f_{c, day}$ are shown in Fig. 7. The effect of the shipping corridor on $N_d$, $r_e$ and $W$ has weakened notably from 2020 onward, acquiring almost half the values of the 2004-2019 period. On average, before 2020 the ship emissions caused an increase in $N_d$ by $7.00 \pm 1.49$ cm$^{-3}$, which weakened to $2.73 \pm 1.50$ cm$^{-3}$ from 2020 onward. Similarly for $r_e$, the average decrease by $0.25 \pm 0.04$ μm due to ship emissions before 2020 weakened to $0.09 \pm 0.04$ μm from 2020 onward. Corresponding average effects on $W$ are $-1.80 \pm 0.47$ g m$^{-2}$ and $-0.99 \pm 0.52$ g m$^{-2}$. In the $f_{c, day}$ case, results are less clear. The across-corridor distribution before 2020 shows some resemblance to the full period distribution (Fig. 3f), although with weaker characteristics. From 2020 onward, no corridor effect is apparent based on the shape of the distribution.

The corridor effects on $N_d$, $r_e$ and $W$ appear statistically significant at the 95% confidence level in both time periods. Similar results were obtained when the differences between the corridor effect profiles before and after 2020 were examined: significant differences were found for $N_d$, $r_e$ and $W$, but not for $f_{c, day}$. These results are summarized in Table 2.

**Table 2: Time series averages (±2σ_std) of the corridor effect on $N_d$, $r_e$, $W$ and $f_{c, day}$ during the periods before 2020, from 2020 onwards, and their difference. Statistically significant effects at the 95% confidence level are shown in bold.**

| | $N_d$ [cm$^{-3}$] | $r_e$ [µm] | $W$ [g m$^{-2}$] | $f_{c, day}$ [%] |
|---|---|---|---|---|
| 2004-2019 | **7.0 ± 6.3** | **-0.25 ± 0.22** | **-1.8 ± 1.6** | -0.02 ± 0.10 |
| 2020-2023 | **2.7 ± 2.4** | **-0.09 ± 0.08** | **-1.0 ± 0.8** | 0.22 ± 0.45 |
| (2020-2023) – (2004-2019) | **-4.3 ± 2.6** | **0.16 ± 0.08** | **0.8 ± 0.5** | 0.24 ± 0.26 |


In the case of $\tau$, since the cubic fit approach does not produce useful results in either of the two periods examined, we analysed the across-corridor profiles of absolute values. As shown in supplementary Figs. S8a and S8b, a significant decrease in $\tau$ is apparent in the period after 2020. This is analyzed further below.

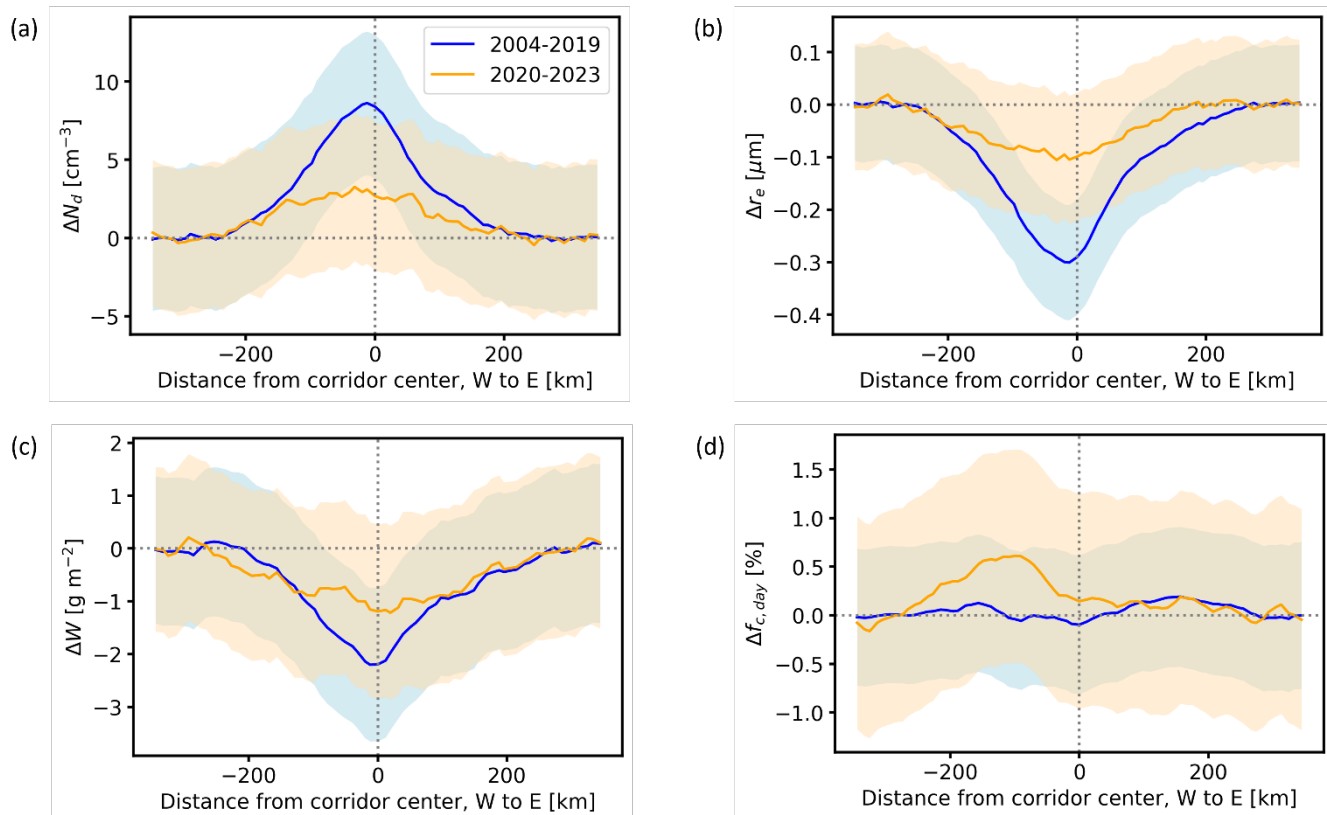

**Figure 7: Corridor effects on $N_d$ (a), $r_e$ (b), $W$ (c) and $f_{c, day}$ (d), in the periods 2004-2019 and 2020-2023, calculated as the differences between the actual average distribution and the no-ship scenarios for these periods. The light-coloured bands show the associated propagated uncertainties (1σ). Dotted vertical lines denote the corridor center. The zero lines are also shown.**

To further examine how the recent changes in ship emissions affected the cloud properties along the corridor compared to less
affected areas, we calculated, for each cloud property and grid cell in the study area, the difference between the averages from
the periods from 2020 onward and before 2020. Results are shown in Fig. 8.

The shipping corridor is clearly visible in the $N_d$ and $r_e$ maps. In the case of $N_d$ it manifests as stronger decreases, compared to
an overall background decrease which is stronger closer to the coast. In the case of $r_e$, it appears as larger increases in the
southern part of the corridor, and smaller increases in the northern part, compared to background decreases. In the maps of $W$
and $f_{c, day}$ differences the corridor is not apparent. Instead, for both variables large decreases appear over the entire area. The
same is true for $\tau$ (supplementary Fig. S8c). As discussed in Sect. 3.2, SST is the main driver for the seasonal variability of $f_{c,}$
$_{day}$, and this is also the case on a longer term. Thus, a possible explanation for this apparent absence of a corridor effect change
during the examined period is that changes due to ship emissions are hidden by a wider and stronger effect of SST changes on
$f_{c, day}$. This scenario is examined in more detail below.

The statistical significance of the differences shown in Fig. 8 was also assessed by comparing the average differences with
their combined uncertainties. In the cases of $N_d$ and $r_e$, differences are significant only in the southeastern part of the region
(Figs. S9a and S9b), while they cover larger parts in the $W$, $f_{c, day}$ (Figs. S9c, S9d), and $\tau$ (Fig. S8d) cases. In no case is the
shipping corridor highlighted from adjacent areas in terms of significance of differences. Similar findings over the region are
also reported in Diamond (2023). The corridor, however, is clearly visible in the $N_d$ and $r_e$ differences maps in Fig. 8, which
is a strong indication that these differences originate in ship emission changes.

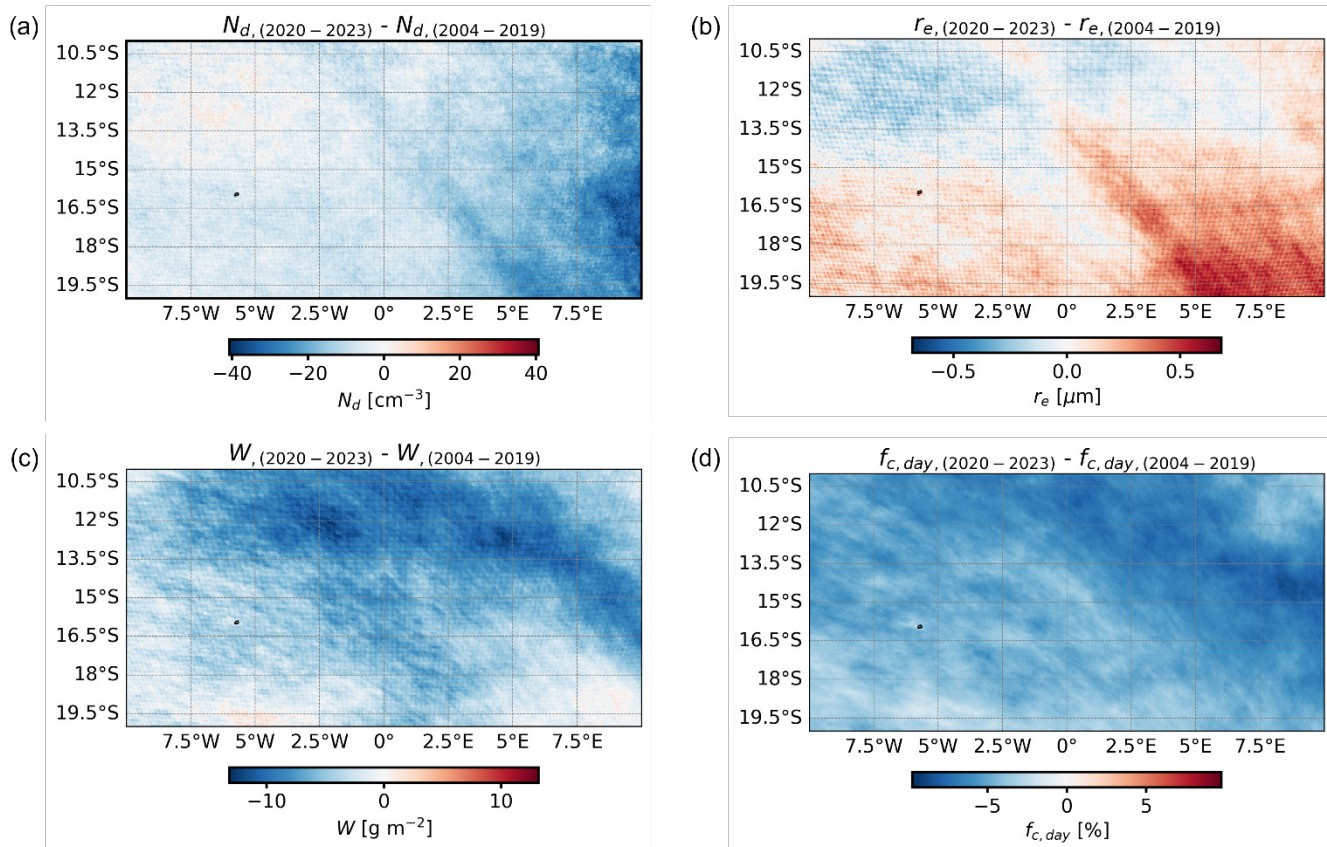

**Figure 8: Maps of $N_d$ (a), liquid $r_e$ (b), $W$ (c) and $f_{c, day}$ (d) differences between the periods after and before 2020. Corresponding uncertainties are shown in supplementary Fig. S9.**

As a further step in the assessment of the long term changes in corridor averages and effects, the time series of monthly average $N_d$, $r_e$, $W$ and $f_{c, day}$ over the corridor and the corresponding corridor effects were also analysed. Figure 9 shows the resulting time series of averages (left column), and evolution of the corridor effects (right column). All time series were deseasonalized and smoothed using a 12-month running average.

Results show a considerable variability in $N_d$ and $r_e$ during the study period (Figs. 9a and c). In more recent years there is a tendency of $N_d$ to decrease and $r_e$ to increase, which may be due to natural variability, anthropogenic causes or a combination thereof. The corridor effect on these two variables is notably weakened in recent years, with values that are not found before 2020 (Figs. 9b and d). Features before 2020 may also be linked to fluctuations in shipping activity. Examining a region in the NE Pacific, Yuan et al., (2022) report dips in the ship track density in 2009-2010 (after-effect of the 2008 financial crisis) and 2014-2016, likely caused by a strong slowdown in the Chinese economy. Abrupt decreases in the corridor effect on $N_d$ in 2010 and 2014 (Fig. 9b) may be associated with these circumstances.

It is also worth noting that $f_{c, day}$ decreases considerably from 2020 onward compared to the period before, with values returning closer to long-term averages in 2023 (Fig. 9g). Similar features also appear in the $W$ time series (Fig. 9e). The corresponding

corridor effects on $f_{c, day}$ and $W$, however, are less notably different from 2020 onward compared to the years before (Fig. 9h
and f). As in the overall changes over the region examined in Fig. 8, the time series of $\tau$ shows very similar characteristics to
those of $f_{c, day}$ and $W$ (Fig. S8b).

As mentioned before, SST is one of the main drivers of cloud variability in marine low-cloud regimes (Andersen et al. 2023),
and this can largely explain these findings. Diamond (2023) also reports low cloud albedo values in 2020-2022 (which are
consistent with the lower $f_{c, day}$ values found here), and suggests that these may be related to unusually warm SST during these
years. Interestingly, Gettelman et al. (2024) suggest that large part of the unusually high SSTs in 2023, especially in the
Northern Hemisphere, can be attributed to the IMO regulations and their effect on clouds, claiming that in 2021-2023 cloud
anomalies were more likely to drive SST anomalies than the other way around. While an analysis of these causation
mechanisms in beyond the scope of this study, an examination of SST data from Aqua MODIS (supplementary Fig. S10)
yields a correlation coefficient R = -0.67 between SST and $f_{c, day}$, confirming their correlation over the study region.

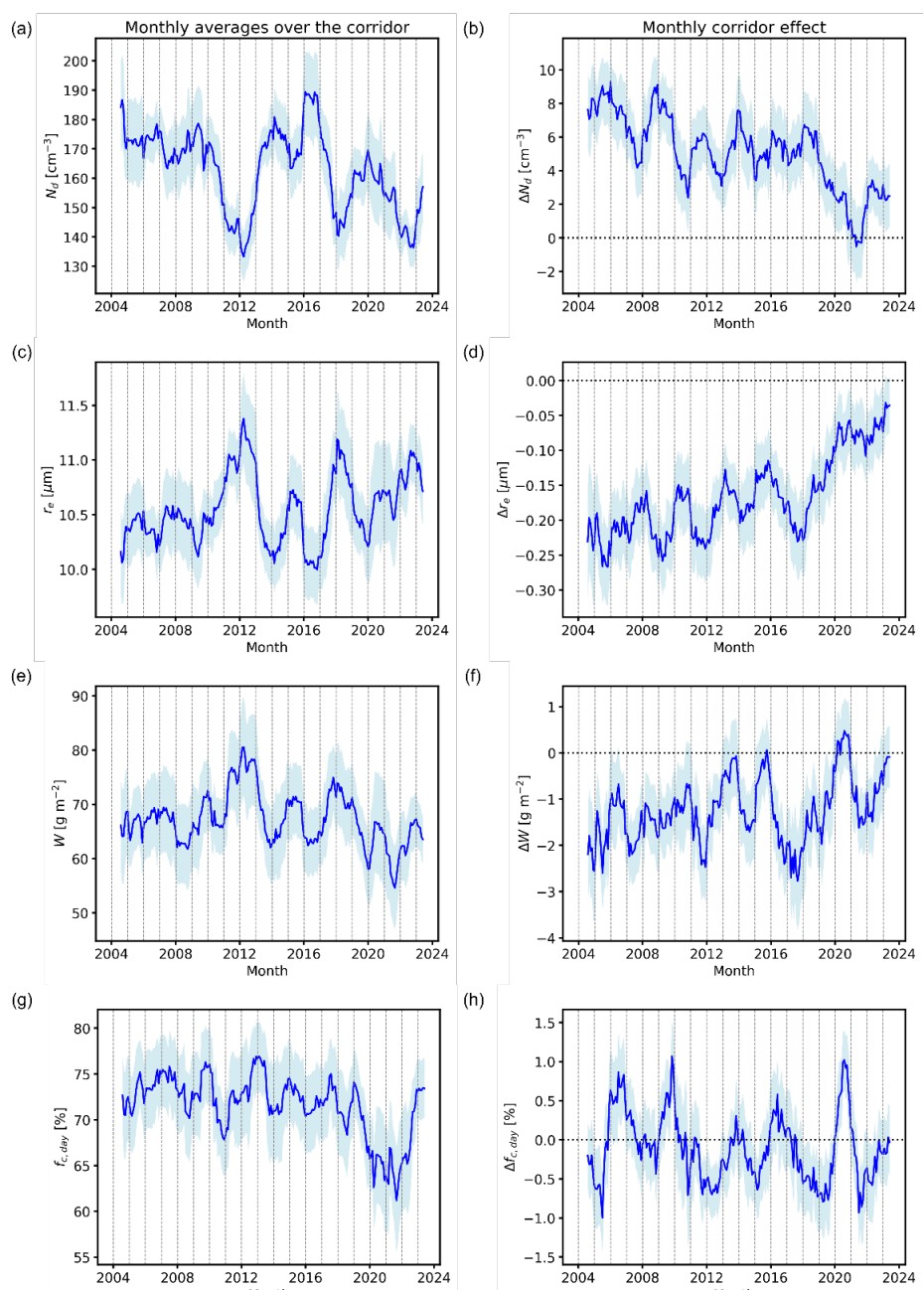


**Figure 9: Time series of monthly average $N_d$ (a), liquid $r_e$ (c), $W$ (e) and $f_{c, day}$ (g) over the shipping corridor in 2004-2023, deseasonalized using a 12-month running average. Corresponding monthly corridor effects are plotted in (b), (d), (f) and (h). The light blue bands show the propagated uncertainty ($1\sigma$) in each case. In all plots, dotted vertical lines denote the beginning of each year. In the corridor effect plots, the zero lines are also shown.**


## 4 Summary and conclusions

In this study we analysed the effect of ship emissions on cloud properties over a busy shipping corridor that crosses the SE Atlantic. The analysis covered the 20-year period 2004-2023, when data from CLAAS-3, the cloud data set based on the geostationary SEVIRI imager, are available. Taking advantage of the CLAAS-3 temporal resolution, the corridor effect on the stratocumulus clouds of the region was quantified on diurnal and seasonal bases, while long term changes were also examined. Results show a clear effect of shipping emissions on the cloud properties associated with the Twomey effect, i.e. an increase in $N_d$ and a decrease in $r_e$. Subsequent adjustments reveal a decrease in $W$, while the cloud fraction changes are more subtle and limited. No clear impact on $\tau$ was found, suggesting an overall minor radiative effect of the shipping emissions, although limitations in our method to detect changes in the corridor cannot be excluded. The effects vary seasonally and diurnally, depending on corresponding regional conditions in the former case and on the cloud thinning during the day in the latter.

In the long term, a weakening of the corridor effect is clearly seen in $N_d$ and $r_e$ from 2020 onward, presumably due to the stricter IMO regulations on sulfur emissions that were implemented at the beginning of that year. Changes in $W$ and the cloud fraction are also detected over the wider region, associated mainly with corresponding SST variations.

Taking advantage of the good alignment between the SE Atlantic shipping corridor and prevailing winds, the approach used here for the quantification of ship emission effects on clouds cannot be directly implemented in other regions and shipping corridors, without prior adjustments in the methodology. The analysis, however, is valuable considering the climatic importance of the extensive SE Atlantic stratocumulus region. It also highlights the great potential of using geostationary-based cloud observations in similar studies, which has not been exploited so far.

## 5 Code availability

All code for the data analysis associated with this study is available at doi.org/10.5281/zenodo.14726844.

## 6 Data availability

CLAAS-3 data were obtained from the CM SAF Web User Interface (https://doi.org/10.5676/EUM_SAF_CM/CLAAS/V003, Meirink et al., 2022). Global shipping traffic density data were obtained from the World Bank Data Catalog (https://datacatalog.worldbank.org/search/dataset/0037580/Global-Shipping-Traffic-Density, Cerdeiro et al., 2020).

## 7 Author contribution

NB and JFM designed the study and performed the analysis. NB wrote the code for the analysis and the first draft of the manuscript. RR and MS provided input on the structure and contents of the manuscript. All authors reviewed and edited the manuscript.

## 8    Competing interests

The authors declare that they have no conflict of interest.

## 9    Acknowledgements

This work was performed within the EUMETSAT CM SAF framework, and all authors acknowledge the financial support of the EUMETSAT member states.

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
