# Peer review of "Analysis of ship emission effects on clouds over the southeastern Atlantic using geostationary satellite observations"

_EGUsphere, 2024_

## Author Response (AR1)

We would like to thank Michael Diamond and the two Anonymous Referees for their insightful reviews. Their corrections and suggestions improved the clarity of the manuscript significantly. Following are our point-by-point responses with the Referee comments in blue font color and italicized. The relevant changes made in the manuscript (page and line numbers) refer to the "track-changes" version of the manuscript.

**Reply to Referee # 1 (M. Diamond)**

*In their manuscript, Benas et al. use a long record of geostationary satellite measurements to investigate the effect of aerosol pollution on low clouds within an isolated shipping corridor in the southeastern Atlantic. Their methods are similar to those employed by my group previously in terms of comparing observations (presumably with shipping effects) with a counterfactual based on cloud properties outside the shipping corridor, and our results are similar in terms of their broad strokes, although there are notable and intriguing differences. Their method is cruder in terms of estimating the counterfactual using a cubic function fit on a reduced-dimension profile centered on the shipping corridor, but more sophisticated in terms of its spatial and temporal resolution. My major concern about the paper is the lack of uncertainty quantification for the counterfactual; although the authors do an admirable job quantifying the uncertainty of the observations, I would argue that we should expect the uncertainty due to estimating the "no ship" counterfactual to be even larger, and more difficult to constrain. Otherwise, my comments are relatively minor. I look forward to seeing an adequately revised manuscript published in ACP. -Michael Diamond*

**Major comments**

*A. Quantifying uncertainty of the counterfactual: The fundamental challenge of quantifying aerosol-cloud interactions in observations is we can't just re-run reality while excluding the aerosol, like we can in a model. The SE Atlantic is such a nice "natural experiment" because the constrained nature of the ship pollution offers us the tantalizing prospect of really being to compare clouds under the same large-scale meteorology differing only with an exogenous aerosol perturbation. As nice as the setup is, however, estimating the "clean" (or at least, non-shipping) cloud field is non-trivial. The cubic fit is a reasonable choice, but I know from experience (albeit with coarser data) that various reasonable-seeming fitting strategies can result in very different answers in terms of the liquid water path (W) and cloud fraction (fc) results. The authors seem to have seen the effect of small variations in methodology as well, in their discussion of shifting the assumption that the non-shipping background starts 150 km instead of 250 km in changing their fc results. I would encourage the authors to do a similar exercise with W; my sense is that moving to 150 km or even 200 km will dramatically shrink the estimated effect magnitude (but not the sign). One suggestion I have in trying to quantify some of the uncertainty in the counterfactual method would be to run the analysis with counterfactual curves fit at 150, 200, 250 (current), and 300 km distances and take the error as the standard deviation from the different fits.*

**Reply:** Thank you for this suggestion. An estimation of the no-ship scenario uncertainty was indeed missing from our analysis and treatment of uncertainties. The uncertainty of the no-ship scenario is now estimated as the standard deviation of five cubic fits, applied by varying the

distance of the assumed background data ranges. The five distances used start from 150 km – 300 km and reach 350 km – 500 km with increments of 50 km, with the range 250 km – 400 km still being used for the quantification of the corridor effect. In each case, the uncertainty of the corridor effect is also updated accordingly, estimated as the combined uncertainty of the actual values and the no-ship scenario. Results show that the cubic fit is a robust assumption in the cases of $N_d$, $r_e$ and $W$. This is not the case with $f_{c, day}$, however, where the no-ship scenario uncertainty confirms the ambiguity in the sign of the corridor effect (see Figs. 2 and 3 in the revised manuscript, and S2 in the revised supplement).

**Changes:** Page 7, lines 198-202, Fig. 2, Fig. 3, lines 242-248, supplementary Fig. S2.

***B.*** *Missing discussion of previous shipping corridor work in introduction: The authors reference Diamond et al. (2020) in their methods and multiple times in comparing results, but do not engage much with the other literature attempting to glean information from shipping corridors instead of individual tracks, including some relevant papers focused on the SE Atlantic as well. Somewhere in the introduction, probably just before current line 69, I would recommend adding a section about the difference between studying individual ship tracks "bottom-up" and shipping corridors "top-down". You should also mention here why the SE Atlantic region is chosen — and why other corridors, such as those investigated by Karsten Peters and colleagues, did not prove conducive to investigating the shipping effect on clouds — and give a summary of what is already known about the corridor.*

*Suggested references:*

*Hu, S., Zhu, Y., Rosenfeld, D., Mao, F., Lu, X., Pan, Z., Zang, L., and Gong, W.: The Dependence of Ship-Polluted Marine Cloud Properties and Radiative Forcing on Background Drop Concentrations, Journal of Geophysical Research: Atmospheres, 126, e2020JD033852, 10.1029/2020jd033852, 2021.*

*Peters, K., Quaas, J., and Graßl, H.: A search for large-scale effects of ship emissions on clouds and radiation in satellite data, Journal of Geophysical Research: Atmospheres, 116, D24205, 10.1029/2011jd016531, 2011.*

*Peters, K., Quaas, J., Stier, P., and Graßl, H.: Processes limiting the emergence of detectable aerosol indirect effects on tropical warm clouds in global aerosol-climate model and satellite data, Tellus B: Chemical and Physical Meteorology, 66, 24054, 10.3402/tellusb.v66.24054, 2014.*

**Reply:** The introduction was rephrased and extended to include these studies and address the points mentioned in this comment.

**Changes:** Page 2, lines 58-64 and page 3, lines 65-75.

Specific comments:

1. *Line 20: Given the uniqueness of the SE Atlantic setup, you might want to soften the statement of generalizability of "for similar analyses" to something more like "studying aerosol-cloud interactions", etc.*

**Reply:** Ok, we have rephrased this statement.

**Changes:** Page 1, lines 21-22.

2. *Line 45: I'm not sure what "typical" means here? I would say they are particularly good examples!*

**Reply:** The term "typical" was replaced by "good".

**Changes:** Page 2, line 47.

3. *Line 183: Related to major comment A above, I would recommend the authors check out Tippett et al. (2024) and update their discussion of Manshausen et al. (2022) accordingly. This reflects the importance and difficulty of estimating a proper counterfactual! I'd also note that the global studies of Wall et al. (2022, 2023) do show globally negative LWP susceptibilities to aerosol.*

   *Tippett, A., Gryspeerdt, E., Manshausen, P., Stier, P., and Smith, T. W. P.: Weak liquid water path response in ship tracks, EGUsphere [preprint], https://doi.org/10.5194/egusphere-2024-1479, 2024.*

   *Wall, C. J., Norris, J. R., Possner, A., McCoy, D. T., McCoy, I. L., and Lutsko, N. J.: Assessing effective radiative forcing from aerosol-cloud interactions over the global ocean, Proc Natl Acad Sci U S A, 119, e2210481119, 10.1073/pnas.2210481119, 2022.*

   *Wall, C. J., Storelvmo, T., and Possner, A.: Global observations of aerosol indirect effects from marine liquid clouds, Atmos. Chem. Phys., 23, 13125-13141, 10.5194/acp-23-13125-2023, 2023.*

**Reply:** We have updated the relevant discussion, including the studies mentioned above. We also note that the two Wall et al. studies are not limited to ship emissions. Because of this, their results may not be directly comparable to those of ship tracks/corridors studies.

**Changes:** Page 10, lines 230-236.

4. *Lines 193-195: Another possibility is simply error in the method! Even if the true effect were zero, we still wouldn't expect to get a result of precisely zero unless the method was absolutely perfect.*

**Reply:** This possibility is now included in the text.

**Changes:** Page 10, line 250.

5. *Line 196: You could also do a quick test to see if you should expect a noticeable perturbation in cloud optical thickness (COT) given your inferred changes in Nd and W. Just from eyeballing Figures 2-3, dln(COT) ~ 1/3 dln(Nd) + 5/6 dln(W) = 1/3(4%) + 5/6(-3%) = -1%. From Fig. S2, I would expect a decrease in COT of ~0.08 to be apparent. The difficulty in obtaining a COT result that fits with your other values could also be a reflection of potential methodological limitations.*

**Reply:** Indeed, based on this estimate, a small decrease in COT would be expected. This is mentioned in the relevant part of the revised manuscript, along with the potential methodological limitation. Namely, that if the corridor does not manifest as a deviation from an otherwise smooth background, no conclusive remarks on its effect can be made.

**Changes:** Page 10, lines 254-259.

6. *Line 196: It might be worth trying to analyze log(COT) instead of COT*

**Reply:** Thank you for the suggestion. With log(COT) available as a separate data set in CLAAS-3, we repeated the analysis, but there are no noticeable differences in the results.

7. *Lines 199-200: I don't see why this should be true. Diurnally or seasonally opposing positive and negative effects would average to zero overall but would be discernible with your method.*

**Reply:** Yes, this is possible. COT is now included in the seasonal and diurnal analysis and results are reported at the end of each section and shown in supplementary figures S4 and S7. However, neither the seasonal nor the diurnal analysis of across-corridor COT profiles reveal any strong corridor-centered perturbation, which would convincingly indicate a corridor effect on COT.

**Changes:** Page 10, line 261, page 11, lines 262-264, page 13, lines 316-318, page 16, lines 368-372, page 20, lines 429-430 and supplementary Figs. S4, S7 and S8.

8. *Line 213: Is this comparison referring to Fig. 9 in Grosvenor et al. (2018)? I don't believe Grosvenor & Wood (2014) provide a seasonal breakdown of the subtropics.*

**Reply:** Indeed, Fig. 9 in Grosvenor et al. (2018) is referred to here. Grosvenor & Wood (2014) is mentioned in that figure, hence the confusion. We have corrected the reference.

**Changes:** Page 12, line 287.

9. *Lines 272-273: However, it should also be noted that geostationary retrievals suffer from diurnally varying biases related to scattering geometry that could be relevant here.*

   *Smalley, K. M., and Lebsock, M. D.: Corrections for Geostationary Cloud Liquid Water Path Using Microwave Imagery, Journal of Atmospheric and Oceanic Technology, 40, 1049-1061, 10.1175/jtech-d-23-0030.1, 2023.*

**Reply:** This limitation of the geostationary retrievals was added. The part on the possible uncertainty between the two MODIS instruments was also rephrased after relevant remarks from the other reviewers.

**Changes:** Page 16, lines 349-356.

10. *Line 276: It's worth noting that negative cloud adjustments at night and positive during the day would be the opposite of what we'd expect from the diurnal cycle of precipitation (maximizing at night) and evaporation (maximizing during the day). See, e.g., Sandu et al. (2008) Figure 7.*

*Sandu, I., Brenguier, J.-L., Geoffroy, O., Thouron, O., and Masson, V.: Aerosol Impacts on the Diurnal Cycle of Marine Stratocumulus, Journal of the Atmospheric Sciences, 65, 2705-2718, 10.1175/2008jas2451.1, 2008.*

**Reply:** We have added this remark and reference in the relevant discussion.

**Changes:** Page 16, lines 359-360.

*11. Lines 281-282: I would not feel safe concluding this…*

**Reply:** This statement stems from examination of shipping corridor and no-ship scenario profiles of $f_c$ on an individual time slot basis. This is probably not clearly shown in the supplementary Fig. S5d (as it is numbered in the revised version), where 24 corridor effect profiles were shown in one plot. In the revised supplement we have included the shipping corridor and no-ship scenario profiles of $f_c$ per time slot in a new figure (supplementary Fig. S6) to further support this conclusion.

**Changes:** Supplementary Fig. S6.

*12. Lines 313-314: Similar conclusions about detectability without using a technique like ML-assisted ship track detection or statistically-generated counterfactual fields were reached by Watson-Parris et al. (2022) and Diamond (2023).*

**Reply:** We have added this remark in the discussion.

**Changes:** Page 18, line 408.

*13. Data availability: I'd encourage the authors to consider making a repository with some processed data needed to reproduce the key figures as well.*

**Reply:** The python code that was used in this study takes unprocessed CLAAS-3 level 3 data as input and is available in doi.org/10.5281/zenodo.14726844. Considering that the total size of the level 3 files analyzed is ~117 GB[1], and that the programs run in seconds to minutes, we consider including intermediate processed data in the study assets unnecessary.

**Changes:** Page 22, lines 468-469.
* * *
[1] This estimation includes the full CLAAS-3 level 3 times series (2004-2023) of product files CFC (cloud fraction variables) and LWP (liquid cloud variables) in their monthly mean (mm) and monthly diurnal (md) averages, i.e. CFCmm, CFCmd, LWPmm and LWPmd.

**Reply to Anonymous Referee #2**

*The study uses geostationary satellite observations over the SE Atlantic to examine the influence of ship emissions on cloud properties. A shipping corridor is defined based on the density of shipping traffic, and cloud properties are sampled both inside the corridor and outside of it, at a safe distance from the influence of the busy shipping corridor. Clouds outside the shipping corridor are considered unperturbed, allowing the influence of ship emissions to be quantified with respect to the perturbed cloud inside the corridor using a cubic fit. The use of geostationary satellite data enables analysis across various time scales, from diurnal variations to seasonal and long-term changes. This complements previous research by Diamond et al. (2020), which used polar satellite data, providing an advancement in this area of study.*

**Major comments**

*A description of the criteria used to filter out clouds that are not liquid low-level clouds is missing. While marine clouds are the most common in the SE Atlantic region, other cloud types are also prevalent. Additionally, it is recommended to exclude cloudy pixels with uncertain retrievals based on lower thresholds of re and τ values (Sourdeval et al., 2016). Were such filters applied here? Given that marine clouds can also exist as broken cloud fields, combined with the coarser spatial resolution of SEVIRI, retrieval uncertainty is likely to be even higher than that of MODIS, for which such filters are frequently applied. Including biased retrievals can affect the averaged values, especially for Nd, due to its high sensitivity to re (Grosvenor et al., 2018). This can introduce a bias in Nd that non-linearly depends on the observed re, which varies between the corridor and the reference regions.*

*Sourdeval, O., C.-Labonnote, L., Baran, A. J., Mülmenstädt, J., and Brogniez, G.: A methodology for simultaneous retrieval of ice and liquid water cloud properties. Part 2: Near-global retrievals and evaluation against A-Train products, Q. J. Roy. Meteorol. Soc., 142, 3063–3081, https://doi.org/10.1002/qj.2889, 2016.*

**Reply:** Separation of liquid from ice clouds in CLAAS-3 is performed in the retrieval of level 2 (instantaneous) cloud phase, as described in Benas et al. (2023). All cloud variables used in the present study come from level 3 data, where they are provided separately for liquid and ice clouds. Thus, successful exclusion of non-liquid clouds in our analysis depends on the performance of the cloud phase retrieval algorithm of CLAAS-3. Relevant validation results, using CALIPSO and MODIS data as reference, show overall good performance of the CLAAS-3 cloud phase algorithm, and very good agreement with the reference data over the SE Atlantic region. These results can be found in the CLAAS-3 validation report, available in Meirink et al. (2022). Regarding filtering of retrievals for thin clouds, no such threshold was applied here. Note, however, that $r_e$ and $τ$ values that lie outside the Nakajima-King LUT in the retrieval process are excluded from level 3 aggregations. These cases usually refer to broken clouds and cloud edges. While we acknowledge that relevant biases in the averaged values cannot be completely excluded, we would not expect variations in these biases between the corridor and adjacent regions. All these points are discussed in the revised manuscript, at the end of Sect. 2.1.

**Changes:** Page 5, lines 134-143.

*The authors chose not to include an analysis of τ because they found no response between the corridor and the reference region, likely due to the cancellation effect between the decrease in W and the increase in Nd. Given that τ is closely related to cloud albedo, does this imply there is no radiative effect from the shipping corridor? Including a radiative perspective could enhance the study's impact by providing further insights into the potential climate effects.*

**Reply:** Based on another reviewer comment, an analysis of $\tau$ is now included also in the study of seasonal and diurnal cycles, since the absence of an effect in the time series averages does not exclude the possibility of seasonally or diurnally opposing positive and negative effects that average to zero but are still discernible with our method. Our findings, however, show that this is not the case: no strong indication of a corridor effect was found in either case (see Figs. S4 and S7 in the revised supplement).

$\tau$ is indeed closely related to cloud albedo, which in turn determines the cloud radiative effect. While the cloud albedo is not available in CLAAS-3, we repeated the analysis for the effective cloud albedo (CAL), which is provided in the CM SAF SurfAce Radiation DAtaset Heliosat (SARAH-3) data record. SARAH-3 consists of seven solar radiation-related parameters, retrieved based on MVIRI and SEVIRI data (Pfeifroth et al., 2024).

Results of the CAL analysis are very similar to those of $\tau$. This is of course expected, but it is worth noting that the two variables are retrieved independently. Due to this similarity, we consider that inclusion of the CAL analysis in the study will not add further insights. The implication that the Referee suggests, however, cannot be readily made based on our results: while the cancellation effect is a possible explanation, another possibility is that, since our methodology limits the discernible effects to those manifesting as deviations from a smooth underlying background, effects of a different shape will not be detected. This possibility is included in the relevant discussion of Sect. 3.1 of the revised manuscript.

For completeness we include here plots from the CAL analysis. Note that a monthly mean diurnal cycle product is not available in SARAH-3, so the diurnal cycle analysis of CAL is omitted. Propagated uncertainties are also not included in the SARAH-3 level 3 data fields, so the corresponding uncertainty bands are also omitted from the plots below.

[Figure]

**Figure 1: (a)** Map of monthly time series average of CAL in 2004-2023. **(b)** Across-corridor average distribution of CAL (blue line); the dotted line shows the no-ship scenario and the grey band is its uncertainty. **(c)** Seasonal variability of the spatially averaged CAL over the study region. **(d)** Map of CAL differences between the periods after and before 2020. **(e)** Time series of monthly average CAL over the shipping corridor in 2004-2023, deseasonalized using a 12-month running average.

Pfeifroth, U., Drücke, J., Kothe, S., Trentmann, J., Schröder, M., and Hollmann, R.: SARAH-3 – satellite-based climate data records of surface solar radiation, Earth Syst. Sci. Data, 16, 5243–5265, https://doi.org/10.5194/essd-16-5243-2024, 2024.

**Changes:** Page 10, lines 256-261. Changes related to the inclusion of the $\tau$ analysis: page 11, lines 262-264, page 13, lines 316-318, page 16, lines 368-372, page 20, lines 429-430 and supplementary Figs. S4, S7 and S8.

**Specific comments**

*Throughout the manuscript, the authors propose hypotheses for their findings that at times seem to be drawn too quickly or are overly speculative, such as in lines 220-223, 270–272 and 279-281. The authors should be more cautious when discussing findings that the current study was not specifically designed to address.*

**Reply:** Lines 220-223: The statement that when using the 3.9 µm channel, the effect of absorbing aerosols on $r_e$ is smaller, compared to using shorter wavelengths, is based on findings from Haywood et al. (2004). To clarify this, we moved the Haywood et al. reference after this statement. We have also rephrased the next sentence as follows: "Since the $N_d$ retrieval depends weakly on $\tau$ and much more strongly on $r_e$ (see e.g. equation (2) in Bennartz and Rausch, 2017), the absorbing aerosol effect on $N_d$ is also expected to be modest compared to retrievals based on smaller wavelengths (e.g. 1.6 µm". We hope this change also clarifies this statement and its origin.

**Changes:** Page 13, lines 296-298.

Lines 270-272: The statement "These results indicate significant uncertainty between the two MODIS instruments", while it refers to results from the studies mentioned before, not our results, is indeed not well supported, since other factors may also play a role in these differences. We have removed it and rephrased accordingly.

**Changes:** Page 16, lines 349-354.

Lines 279-281: We don't see a speculative or unsupported hypothesis here. In fact, we state that "no concrete conclusion can be drawn regarding the effect of ship emissions on $f_c$ during the day". The next sentence ("It can be safely concluded, however, that during night the shipping corridor exerts a negative effect on $f_c$") is based on examination of difference profiles at specific night time slots, shown in supplementary Fig. S4d (Fig. S5d in the revised supplement), which includes all 24 time slots. In Fig. S6 of the revised supplement we have included all 24 time slot profiles of $f_c$ across the corridor, with their no-ship scenarios. These plots show more clearly the corridor effect on the profiles during night, supporting our conclusion.

**Changes:** Page 16, lines 366-367, supplementary Fig. S6.

*The differences found in the cloud properties are small, yet the axis range set in the plots make them appear more significant. This could be somewhat misleading. Why not present the results also as relative changes?*

**Reply:** The y-axis ranges in all plots where selected to include the full range of absolute or difference values, and corresponding uncertainties, in the corridor and its surroundings. Following the Referee's suggestion, we have included values of the average absolute and relative changes (± corresponding uncertainties) in the differences plots of Figs. 2e, 2f, 3e and 3f. We agree that inclusion of relative differences (in %) will give a better impression on the significance of these changes.

**Changes:** Figs. 2 and 3

*Line 172: re should be $r_e$.*

**Reply:** Corrected.

**Changes:** Page 9, line 220.

*Line 252-256: Can you provide a reference for why cloud thinning would lead to a smaller re? It depends on homogeneous versus inhomogeneous mixing.*

**Reply:** The Referee is right. Cloud thinning caused by entrainment of dry air would lead to a smaller $r_e$ under homogeneous mixing conditions. When the mixing is inhomogeneous, droplet size does not change significantly (Yeom et al. 2023).

Our interpretation of the decreasing $r_e$ during the day stems from the fact that in adiabatic stratocumulus clouds particle size typically increases with height. Thus, when the clouds are (geometrically) thinner, the droplets at the top will be smaller, leading to a smaller retrieved $r_e$ (Brenguier et al. 2000).

Brenguier, J.-L., Pawlowska, H., Schuller, L., Preusker, R., Fischer, J., and Fouquart, Y.: Radiative properties of boundary layer clouds: Droplet effective radius versus number concentration, J. Atmos. Sci., 57, 803-821, https://doi.org/10.1175/1520-0469(2000)057<0803:RPOBLC>2.0.CO;2, 2000.

Yeom, J. M., Helman, I., Prabhakaran, P., Anderson, J. C., Yang, F., Shaw, R. A., and Cantrell, W.: Cloud microphysical response to entrainment and mixing is locally inhomogeneous and globally homogeneous: Evidence from the lab, P. Natl. Acad. Sci. USA, 120, e2307354120, https://doi.org/10.1073/pnas.230735412, 2023.

**Changes:** Page 14, line 332 and page 15, line 333.

*Line 256: This sentence in not clear. Nd can be calculated from Terra, why do you say "since no Terra Nd product is available"?*

**Reply:** Here we refer to equation (9) in Diamond et al. (2020). As also mentioned there, "no published Terra $N_d$ product is currently available". This is still the case for the latest MODIS C061 Cloud Product. But $N_d$ can indeed be calculated from Terra, so we have rephrased this part for clarity.

**Changes:** Page 15, line 337.

*Line 274: Cloud fraction depends on the thresholds used to distinguish between cloudy and clear pixels. Perhaps the different threshold at night and day is related to the corridor effect changing between night and day?*

**Reply:** While this is a possibility that cannot be fully excluded, we don't see why it would happen. The corridor effect is estimated based on retrievals from within the corridor and from surrounding areas. These in- and outside corridor areas will be similarly affected by different thresholds at night and day. In other words, the retrieval thresholds should affect the cloud mask differently inside and outside the corridor at night and day, in order to have an impact on the diurnal variation of the corridor effect on $f_c$. This appears unlikely.

It should be added here that the CLAAS-3 validation of $f_c$ against CALIOP data results in very good scores, and the day and night retrievals appear overall consistent, with only a few irregularities in twilight conditions (defined as $75° < \vartheta_0 < 95°$). These details can be found in the CLAAS-3 validation report, available in https://doi.org/10.5676/EUM_SAF_CM/CLAAS/V003.

*Line 311: I don't see a statistical significance test (S5 that you refer to shows maps of uncertainties). For the trend analysis related to IMO regulations, performing a significance test between the two time periods would be useful for quantifying the differences.*

**Reply:** A statistical significance test between the two time periods was performed by comparing the absolute difference between the two periods with their combined uncertainty. It was discussed in lines 311-314 of the discussion paper, without including the relevant maps, as the Referee points. Results of the test are now included in Fig. S9 of the revised supplement, where we use a color scale for the significant cases and a grey scale for the non-significant ones. We have also revised the relevant discussion in the manuscript, adding that, while the shipping corridor is not highlighted from the surrounding regions in the statistical significance test results, the fact that it is clearly visible in the $N_d$ and $r_e$ differences maps in Fig. 8 is a strong indication that these differences originate in ship emission changes.

**Changes:** Page 18, lines 408-410, supplementary Fig. S9.

*Line 313: What is CFC?*

**Reply:** CFC is the Cloud Fractional Coverage, referred to as $f_{c, day}$ throughout the manuscript. We have corrected it.

**Changes:** Page 18, line 407.

*Lines 360-363: This should be included earlier, in the methodology section.*

**Reply:** Indeed. In the revised manuscript it is moved at the beginning of Sect. 2.3, and the summary and conclusions section are adjusted accordingly.

**Changes:** Page 6, lines 165-167, page 22, lines 459-463.

**Reply to Anonymous Referee #3**

*The manuscript documents a SEVIRI based analysis of the potential effects of ship emissions on low-cloud properties. The uniqueness of this analysis is that the authors make use of geostationary retrievals. However, it is hard to discern what is new, especially from the abstract, as the findings are comparable to those described in Diamond et al. (2020). While some differences between Benas et al and other studies are attributed to the use of Aqua MODIS and Terra MODIS and plausible calibration issues (which is far from being certain), the main findings of Benas et al. are a corroboration of Diamond et al and other studies. The novel result of this study is the analysis of the diurnal cycle, but the ship track signal is weak, except for cloud fraction. In addition, the trend analysis for the entire SEVIRI record is not conclusive, and the following sentence in the abstract is not supported by the evidence "The long-term analysis reveals a weakening of the shipping corridor effect on Nd and re presumably following the International Maritime Organization's 2020 stricter regulations on sulfur emissions…" I appreciate the effort but, again, the authors need to more clearly state what is new of their study.*

**Reply:** We have edited part of the abstract and main text, to highlight novel aspects of this study. We agree that a large part of our findings corroborate Diamond et al and findings from other studies, but we also claim that the novel aspects in our study are not limited to the diurnal cycle analysis; the full-year seasonal analysis and the time series analysis, both based on monthly resolved data, are also novel in their temporal coverage and resolution. Regarding the diurnal cycle of the corridor effect: we agree that the magnitude of the diurnal cycle is not large for $N_d$, $r_e$ and $W$, while ambiguities appear for cloud fraction. This is discussed in the text, and shown in supplementary Fig. S4 (S5 in the revised version). However, we would argue that finding a weak diurnal variation is also a result worth showing and discussing.

Regarding the long-term analysis, Figs. 9b and 9d clearly show a previously unseen weakening of the corridor effect on $N_d$ and $r_e$ after 2020, so we don't see why the relevant sentence in the abstract is not supported. In fact, we attempted to carefully phrase this sentence in order not to conclusively attribute this change to the IMO regulations.

The novel aspects of this study can be summarized as follows:

- use of cloud property retrievals from geostationary observations for the first time in a study of ship emission effects on clouds (and the ensuing diurnal cycle analysis)
- full seasonal cycle analysis
- high temporal resolution in the time series analysis.

As mentioned above, we try to emphasize more clearly these new aspects in various parts of the revised manuscript. Please note, however, that we also consider corroboration of previous findings, based on a different sensor and methodology, as a valuable outcome of our analysis.

**Changes:** Page 1, line 10, page 4, line 99, page 16, lines 354-355.

**Other comments**

*Difference between SEVIRI and MODIS: While I agree with the authors about issues with some channels for Terra MODIS, I am not aware of any quality degradation of the retrievals for Terra MODIS relative to Aqua (line 270-271 should be rephrased because the results in Benas et al. do not indicate significant uncertainties between the 2 MODIS instruments). Moreover, sensor issues can also be invoked for SEVIRI, as more than 1 SEVIRI instrument provided the data for this study. While I would agree with the authors that SEVIRI is a stable sensor, I am more interested in learning about pixel resolution*

*differences between MODIS and SEVIRI and how they would affect the findings. More specifically, what would be the impact of the 3-km (nadir) pixel resolution of SEVIRI versus MODIS (1 km). This could be the single most important difference between this study and Yuan et al./Diamond et al. A coarser resolution would certainly impact the cloud mask identification, and cloud retrievals (I would expect larger droplet effective radii and smaller optical depth as the pixel resolution is degraded). Is the 3-km pixel resolution sufficient for detecting ship tracks? Possibly, the pixel degradation with viewing zenith angle is minimized for the study region (which is good); however, there is no quantitative description of a solar zenith angle threshold.*

**Reply:** We agree with the Referee that the statement in lines 270-272 is not a result of our analysis; it was meant to highlight the non-conclusive results of the MODIS-based studies mentioned in lines 266 and 270. However, other factors may also play a role in the reported differences between these studies, so the statement was removed, and the sentence was rephrased accordingly.

**Changes:** Page 16, lines 351-356.

A coarser resolution will indeed lead to decreased $\tau$ and increased $r_e$ values; this is the result of an increased probability of pixels containing a mix of cloudy and clear sky, which will lead to lower reflectance values in both the visible and SWIR channels. This can lead to biases with corresponding MODIS results, due to the higher spatial resolution of the latter. It is not expected, however, to affect the corridor effect significantly: the effect is estimated based on retrievals from the corridor and surrounding areas, which are similarly affected (biased) in SEVIRI compared to MODIS.

The effect of the SEVIRI coarser resolution (compared to MODIS) on the results cannot be completely disentangled from other differences, based on the current setups of this study and the one by Diamond et al. (2020). The definition of the shipping corridor and the methodology to estimate its effect on clouds are probably the two major differences of these other differences. While we consider the comparisons with MODIS a valuable part of our analysis, and our study was largely inspired by the MODIS-based Diamond et al. (2020) study, a full analysis of the mentioned differences would be beyond scope. However, in Sect. 3.1 we have added a relevant discussion on possible differences due to the different spatial resolution.

**Changes:** Page 11, lines 265-274.

Additionally, we attempt here a comparison with part of the results presented in Table 1 of Diamond et al. (2020), by adjusting our temporal coverage and months examined to match those of Diamond et al. (2020) as closely as possible (2004-2015 and September-October-November, respectively). As mentioned above, we only consider the SEVIRI time slots closest to the Terra and Aqua overpasses. The mean corridor value of $\tau$ appears consistently lower in CLAAS-3 compared to MODIS/Terra and Aqua values, as would be expected due to a coarser resolution. This is not the case with $r_e$, where results are similar for Terra and lower in CLAAS-3 compared to Aqua. Differences increase in the cases of $W$ and especially $N_d$, for which additional assumptions/filters are used in the MODIS retrievals compared to CLAAS-3 (Bennartz and Rausch, 2017). Overall, and considering the factors that can lead to deviations, the results of the two studies, including corridor effect values, appear consistent. Note that corridor effect values of $\tau$, although reported here, should probably not be considered meaningful, as explained in our response to the next comment.

| | | MODIS/Terra (D20) | SEVIRI CLAAS-3 at Terra overpass | MODIS/Aqua (D20) | SEVIRI CLAAS-3 at Aqua overpass |
|---|---|---|---|---|---|
| $r_e$ [μm] | Mean Ship value | 10.83 | 10.98 | 11.41 | 10.26 |

| | | | | | |
|---|---|---|---|---|---|
| | Absolute Ship – NoShip difference | -0.28 | -0.31 | -0.29 | -0.27 |
| | Relative Ship - NoShip difference (%) | -2.61 | -2.86 | -2.52 | -2.59 |
| $\tau$ [-] | Mean Ship value | 11.07 | 9.93 | 8.73 | 7.13 |
| | Absolute Ship - NoShip difference | 0.24 | 0.23 | 0.05 | 0.09 |
| | Relative Ship – NoShip difference (%) | 2.13 | 2.36 | 0.58 | 1.23 |
| $W$ [g m$^{-2}$] | Mean Ship value | 85.23 | 77.19 | 66.08 | 51.70 |
| | Absolute Ship – NoShip difference | -0.49 | -0.56 | -1.32 | -0.80 |
| | Relative Ship - NoShip difference (%) | -0.57 | -0.73 | -2.00 | -1.55 |
| $N_d$ [cm$^{-3}$] | Mean Ship value | - | 125.24 | 93.25 | 131.78 |
| | Absolute Ship - NoShip difference | - | 9.08 | 4.87 | 8.40 |
| | Relative Ship - NoShip difference (%) | - | 7.25 | 5.22 | 6.38 |

The question on the sufficiency of the SEVIRI 3 km resolution for detecting (individual) ship tracks is an interesting one, which we plan to investigate in the future. The answer, however, would not affect the results of the present study. Compared to individual ship tracks, the shipping corridor studied here is characterized by a continuous provision of emitted aerosols by many ships crossing the region, while their movement in both directions and spread in space result in a wider affected area. These conditions lead to effects on clouds that are discernible from SEVIRI.

As the Referee mentions, this region is favorable in terms of viewing conditions, with low viewing zenith angles and minimum expected pixel degradation. Regarding the solar zenith angles, the CLAAS-3 algorithm retrieves $\tau$ and $r_e$ for values of $\vartheta_0 < 75°$, which is used as threshold to define day light conditions. This is included in Sect. 2.1 of the revised manuscript.

**Changes:** Page 5, lines 132-133.

*Statistical analysis: Considering the tiny changes in microphysical properties, a robust statistical method for testing the hypothesis is essential and should be highlighted throughout the article. I am also somewhat concerned about the methodology to construct Fig 3c-f. Particularly for cloud fraction (f_c, Fig 3f), I don't understand why the pattern is undulating. This makes me speculate that the methodology is not ideal for identifying the effect of ship tracks because cloud spatial variability is likely dominating the signal. The same comment applies to optical depth (tau) in the supplement; the tau pattern does not make sense. If tau were unperturbed by ship emission, deltatau as a function of the corridor distance should be a flat curve, right?*

**Reply:** Throughout the manuscript we report propagated uncertainties along with all variables. These uncertainties allow evaluating whether differences between variables – inside/outside the corridor or before/after 2020 – are significant.

We do not claim that the methodology we use for identifying the effect of the shipping corridor on cloud properties is ideal. It assumes that the corridor manifests as a deviation from an otherwise smooth background, and this is not always the case. This is stated in Sect. 2.3 of the revised

manuscript. When this assumption is not valid, results are not meaningful. This is clearly true for $\tau$, but also for parts of the seasonal and diurnal analyses. For this reason, we always examine individual profiles of the estimated corridor effects (shown in Figs. S4, S5 and S6 in the revised supplement), when reporting average values. Cloud spatial variability may indeed dominate the corridor signal in these cases. In most cases, however, this approach provides meaningful results. For example, the corridor signal is prominent in $N_d$, $r_e$ and $W$. Based on another review comment, we estimate the no-ship scenario uncertainties by repeating the calculations while varying the distance of the assumed unaffected ranges from the corridor center. This revised approach shows that our method is robust for calculating the corridor effect on these three variables, while it highlights its limitations in the case of $f_c$ (see e.g. Figs. S2 and S6 in the revised supplement). As mentioned above, in the case of $\tau$ the underlying smoothness assumption does not hold, so the results are not meaningful. $\Delta\tau$ is not shown, so we are not sure what the Reviewer refers to. Based on our methodology, if $\tau$ were unperturbed, and its across-corridor distribution were (perfectly) smooth, $\Delta\tau$ would be zero.

**Changes:** Page 6, line 176, page 7, line 177, supplementary Figs. S2 and S6.

*Last paragraph of page 7 should be discussed in section 2.3*

**Reply:** Section 2.3 is meant to discuss the method for estimating the shipping corridor effect, not the results. Thus, our references to Fig. 2 in this section were confusing. In the revised manuscript we have removed all references to results from Sect. 2.3.

**Changes:** Page 7, lines 179-180, 182-183.

*Equation 1: I am not sure why the combination of 2 dissimilar variables (standard deviation and retrieval uncertainty) can be used to produce the uncertainty of averaged data. At least, it does not seem to be mathematically correct.*

**Reply:** This equation is a short version of Eq. (5) in Stengel et al. (2017). It considers both the retrieval uncertainty, which is an output of the CPP algorithm based on input data uncertainties, and natural variability. According to Eq. (5) in Stengel et al. (2017):

$$\sigma_{\langle x \rangle}^2 = \frac{1}{N}\sigma_{natural}^2 + c\langle \sigma_i \rangle^2 + (1-c)\frac{1}{N}\langle \sigma_i^2 \rangle$$

where $\sigma_{natural}$ is the natural variability, $\langle \sigma_i \rangle = \frac{1}{N}\sum_{i=1}^{N}(\sigma_i)$ and $\langle \sigma_i^2 \rangle = \frac{1}{N}\sum_{i=1}^{N}(\sigma_i^2)$. Given that $\sigma_{natural}^2 = \sigma_{std}^2 - (1-c)\langle \sigma_i^2 \rangle$ (Eq. (4) in Stengel et al. (2017)), we get:

$$\sigma_{\langle x \rangle}^2 = \frac{1}{N}\sigma_{std}^2 - \frac{1}{N}(1-c)\langle \sigma_i^2 \rangle + c\langle \sigma_i \rangle^2 + (1-c)\frac{1}{N}\langle \sigma_i^2 \rangle = \frac{1}{N}\sigma_{std}^2 + c\langle \sigma_i \rangle^2$$

*Line 171 You mean "determine" instead of "simulate"*

**Reply:** Yes, replaced.

**Changes:** Page 9, line 219.

*Page 9 "This result suggests that two opposite tendencies, namely an increase in $\tau$ due to the Twomey effect and a decrease due to the decreasing $W$ cancel each other out." This is an overinterpretation of*

*the satellite data because from a remote sensing perspective droplet effective radius is the key variable, not LWP (LWP is indirectly estimated from effective radius and optical depth).*

**Reply:** Indeed, $W$ is estimated indirectly from $r_e$ and $\tau$. We don't see, however, why this statement cannot be possible, due to this retrieval feature. Our purpose was to present it as a possibility, not as an outcome of our analysis. We clarify this point in the revised manuscript.

**Changes:** Page 10. Lines 257-261.

*Line 238: "it is difficult to draw any conclusion on the shipping corridor effect on fc, day." Is there a sentence missing here?*

**Reply:** No, "$f_{c,\,day}$" denotes the daytime cloud fraction. The confusion was likely caused by an unfortunate line break between 'c' and 'day'.

*Introduction: Lines 22-43. Future readers already know this. For brevity, the authors should*

**Reply:** This sentence is incomplete. Does the Referee mean to say that this part of the introduction is redundant? We agree that this is common knowledge in our field, but we also think it is good to start such a study with a broader overview, which serves the purpose of placing the specific questions addressed here in a wider context.

---

## Author Response (AR2)

We would like to thank Michael Diamond and the two Anonymous Referees for reviewing the revised manuscript. Following are our point-by-point responses to their latest comments. Line numbers in our replies refer to the "track-changes" version of the manuscript.

**Reply to Referee # 1 (M. Diamond)**

*The authors have done a nice job responding to the reviewer comments and the resulting manuscript is in very good shape. I have a few remaining comments that I hope the authors will consider before publication. -Michael Diamond*

*1. For the displayed uncertainty, please specify the confidence interval. (I.e., is this one propagated standard error or two?)*

**Reply**: The displayed uncertainties correspond to one standard error. This is now clarified in the captions of Figs. 2, 4, 7, 9, S1, S2, S3, S6, S7, S9.

*2. Lines 216-218:*

*a) The Tippett results are specifically about the "invisible" ship tracks method utilizing air mass tracking, not all track methods (like those picking out visibly identified tracks). I'd suggest adding something like: "More recently, Tippett et al. (2024) showed that the W response onto aerosol perturbations from ship emissions is weak on average, after correcting for biases in prior research <based on tracking ship-affected air masses (Meinshausen et al., 2022, 2023)>, related to correlations between wind and cloud properties." Probably also worth citing Toll et al. (2019).*

**Reply**: We rephrased this sentence as suggested (lines 226-228).

*b) I'd recommend briefly discussing the visible track literature as well given the scope of the statements you're making. There are numerous possibilities, but the most comprehensive for visible tracks globally would be: Toll, V., Christensen, M., Quaas, J., and Bellouin, N.: Weak average liquid-cloud-water response to anthropogenic aerosols, Nature, 572, 51-55, 10.1038/s41586-019-1423-9, 2019.*

**Reply**: The main conclusion of the Toll et al. (2019) study regarding the effect of visible ship tracks on *W* was added in this discussion (lines 228-299).

**Reply to Anonymous Referee #2**

*The authors addressed most of my concerns. However there are a few important lingering aspects that were not fully resolved in the revised manuscript.*

*1. Abstract: It is not sufficient to say that the contribution of this manuscript is the use of geostationary cloud retrievals while listing findings that are also reported in other articles. Again, the abstract needs to document the novel contribution of this paper. In reply you did mention some new results, which should be added to the abstract.*

**Reply**: We have included these additional novel contributions in the revised abstract.

*2. I still do not see a statistical analysis that tests the statistical significance of the magnitude change over the shipping corridor. This analysis of the signal significance is different from the uncertainty analysis they provide. Basically, a statistical test needs to be applied to demonstrate that the changes over the corridor are statistically different from zero. Standard procedures can be found in statistical and climate analysis books (e.g. Wilks et al., von Storch & Zwiers).*

**Reply**: A statistical significance analysis of the corridor effect, separately from the uncertainty propagation, was indeed missing. We have now included it in the time series average, monthly effects, and the profiles of the corridor effects before and after 2020 (lines 171-173, 206-208, 216-217, 232, 268-269, 311-313, 326-328, 385-387 and Tables 1 and 2).

*Other comments*

*Line 133: daytime instead of day-only?*

**Reply**: We changed day-only to daytime (line 121).

*Line 233: Strong inversion would make cloud -op entrainment less likely, so it is not easy to reconcile the idea of strong inversion and enhanced entrainment, it is conflicting.*

**Reply**: The referee is right, that our phrasing is inaccurate. The first part of the sentence is a general characteristic of stratocumulus clouds, namely that they have a strong inversion at the top. The second part of the sentence describes the consequence of an increase in droplet number concentration: increased entrainment and decreasing liquid water path. We removed the first part of the sentence and slightly reformulated the second part (lines 221-222).

*Line 252: "driven by a modification of the air circulation across the corridor", what physical mechanism could explain this process?*

**Reply**: This phrase was added to clarify how the presence of the corridor could have an effect on its sides. Across-corridor circulation patterns have been described before for different mechanisms (Wang and Feingold, 2009). In our case, we do not want to speculate on mechanisms that cannot be justified based on our analysis. Therefore, this phrase is removed (line 244).

*Figure 4: is there an annual cycle in the shipping activity?*

**Reply**: Unfortunately, we do not have seasonally resolved data related to shipping activity to answer this question.

*Line 331-332: Changes during the daytime are strongly modulated by shortwave radiative heating. Contrary to the explanation provided by the authors, entrainment rate peaks during nighttime.*

**Reply**: Both our first attempt to give a plausible explanation of the slight decrease in $r_e$ during the day, and our second attempt to clarify it, caused remarks by two Referees pointing to possible misconceptions and inconsistencies and highlighting the complexity of the issue. Given that this explanation is not crucial for the main findings of the study, and to avoid possible inaccuracies, we decided to remove this sentence (lines 331-334).

*Line 348" This contrasts with"*

**Reply**: Corrected (line 349).

*Line 354: "however the latter…" what is the latter. It is not clear from the sentence.*

**Reply**: With "the latter" we mean observations in more frequent time intervals, such as CLAAS-3. We have replaced "the latter" with "diurnal observations" to clarify this (line 354).

**Reply to Anonymous Referee #3**

*The authors have made a thorough revision of the manuscript. I believe this work represents an advancement in the field with respect to the analysis of the diurnal cycle using geostationary data.*

*In my initial review, I pointed out that the lack of response in τ is intriguing, as it might suggest, for example, that the Twomey and LWP effects cancel each other out. If this is the case, it is particularly interesting, as radiative forcing is ultimately what matters. In the revision the authors addressed this point by performing an analysis of τ as well as of the effective cloud albedo. The results for cloud albedo were similar to those for τ, as expected. What does this imply for the radiative impact of the shipping corridor? I think that this point should be emphasized in the discussion.*

**Reply**: The absence of any apparent impact on τ and the consequent implication on the radiative impact are now mentioned in the summary and conclusions section, as well as in the abstract (lines 15-16 and 460-461). However, given the discussed limitations in our methodology, we refrain from definitely concluding that there is indeed no impact on these parameters (see also our reply to the comment below).

*One more point on that regard is that the authors suggest in the revised manuscript that the absence of an apparent corridor effect in τ could indicate a limitation of their methodology. If so, I wonder how does this limitation apply to the rest of their results.*

**Reply**: We imply here that our method will not work well if the corridor effect does not manifest over a smooth (across-corridor) background. In this case, while the ship emissions may still have an effect on τ, this effect cannot be quantified using our methodology. When a smooth background is present, we are confident that our method provides meaningful results on the quantification of the corridor effect.

*Please note that a correction is needed in the caption of Figure S4 regarding τ. Also, in 372 of the track changes you should add S before Figs. 7.*

**Reply**: We corrected both.

*I also want to comment on the authors' response to my remark from the first review: "Line 252-256: Can you provide a reference for why cloud thinning would lead to a smaller re?." - Thin Sc clouds do not necessarily indicate limited cloud growth, they could also result from cloud dissipation. In such cases, the cloud particle size does not necessarily increase with height, and the clouds are not adiabatic. This might affect your interpretation.*

**Reply**: Thank you for this remark. The intension of that sentence in the first submission ("Liquid $r_e$ decreases slightly in the morning (Fig. 5c), probably due to the overall thinning of clouds") was to provide a plausible explanation on the slightly decreasing $r_e$ during the day. Two Referees pointed deficiencies and possible misconceptions in both this initial phrasing and our attempt to clarify it, highlighting the complexities of the underlying mechanisms. Given that an explanation of this plot is not crucial for our main findings, we decided to remove this sentence (lines 331-334).